# It's not all abundance: Detectability and accessibility of food also explain breeding investment in long-lived marine animals

Enric Real[1,2]*, Daniel Oro[1,3], Albert Bertolero[4], José Manuel Igual[1], Ana Sanz-Aguilar[1,5], Meritxell Genovart[1,3], Manuel Hidalgo[2], Giacomo Tavecchia[1]

1 Animal Demography and Ecology Unit, Instituto Mediterráneo de Estudios Avanzados, Esporles, Spain, 2 Instituto Español de Oceanografía, Centre Oceanográfico de Baleares, Palma, Spain, 3 Centre d'Estudis Avançats de Blanes, CEAB (CSIC), Blanes, Spain, 4 Associació Ornitològica Picampall de les Terres de l'Ebre, Amposta, Spain, 5 Applied Zoology and Conservation Group, University of Balearic Islands, Palma, Spain

* enrique.real@ieo.es

## Abstract

Large-scale climatic indices are extensively used as predictors of ecological processes, but the mechanisms and the spatio-temporal scales at which climatic indices influence these processes are often speculative. Here, we use long-term data to evaluate how a measure of individual breeding investment (the egg volume) of three long-lived and long-distance-migrating seabirds is influenced by i) a large-scale climatic index (the North Atlantic Oscillation) and ii) local-scale variables (food abundance, foraging conditions, and competition). Winter values of the North Atlantic Oscillation did not correlate with local-scale variables measured in spring, but surprisingly, both had a high predictive power of the temporal variability of the egg volume in the three study species, even though they have different life-history strategies. The importance of the winter North Atlantic Oscillation suggests *carry-over* effects of winter conditions on subsequent breeding investment. Interestingly, the most important local-scale variables measured in spring were associated with food detectability (foraging conditions) and the factors influencing its accessibility (foraging conditions and competition by density-dependence). Large-scale climatic indices may work better as predictors of foraging conditions when organisms perform long distance migrations, while local-scale variables are more appropriate when foraging areas are more restricted (e.g. during the breeding season). Contrary to what is commonly assumed, food abundance does not directly translate into food intake and its detectability and accessibility should be considered in the study of food-related ecological processes.

## Introduction

An important challenge in the study of population fluctuations is to reveal the link between demographic parameters and climatic variables, mediated by their influence on foraging resources [1–3]. It is difficult, however, to single out the effect of a single climatic variable on

**Data availability statement:** All data are available through the following link: https://digital.csic.es/handle/10261/259323.

**Funding:** Funds were supplied by grants CGL2013-42203-R and CGL2017-85210-P (MCIU/AEI/FEDER, UE) respectively. The study also received funding from the European Commission's Horizon 2020 Research and Innovation Program under Grant Agreement no. 634495 for the project Science, Technology, and Society Initiative to Minimize Unwanted Catches in European Fisheries (MINOUW). MG was partially funded by Govern Balear. ASA was supported by a Ramón y Cajal contract (RYC-2017- 22796). The funders had no role in study design, data collection and analysis, decision to publish, or preparation of the manuscript.

**Competing interests:** The authors have declared that no competing interests exist.

a given biological system because variables can act directly [4], indirectly through multiple paths [5], alone [6] or in combination with others [7]. For this reason, large-scale climatic indices are often preferred as predictors of ecological processes than local variables, because they integrate environmental changes over different temporal and spatial scales [8–11].

The North Atlantic Oscillation (NAO) and the Southern Oscillation indices (SOI), for example, have been used in many studies as ecological predictors in both terrestrial and marine ecosystems (see [12, 13] and references therein). In marine ecosystems, the winter NAO (hereafter wNAO) is known to influence demographic parameters, such as reproductive success and survival in long-lived and long-ranging top predators [14–16]. It is often assumed that the influence of these large-scale climatic indices occurs *via* their influence on local climatic variables (see [12, 17] and references therein) and/or *via* the indirect effects on local food abundance (see [9, 12, 18] and references therein). However, in many cases, the mechanisms and the spatio-temporal scales through which these climatic indices influence demographic parameters remain largely unexplained or speculative [8, 19].

The combined use of large-scale climatic indices and local-scale variables should reveal their relative role and describe the mechanisms and the spatio-temporal scale at which they influence these demographic parameters [20–23]. The Scopoli's shearwater *Calonectris diomedea*, the Sandwich tern *Thalasseus sandvicensis*, and the Audouin's gull *Ichthyaetus audouinii* are three examples of long-lived and long-distance migratory marine top predators. These three species breed sympatrically in the western Mediterranean, have different life-histories and feed at different depths in the water column: gulls are surface feeders, terns make short diving plunges and shearwaters perform much deeper and longer foraging dives. Furthermore, terns and gulls can modulate clutch size depending on environmental conditions, whereas shearwaters lay a single egg. Despite these differences, foraging areas of the three species overlap both during the breeding [24–26] and wintering seasons [27, 28], in particular along the western coasts of Africa (Figs 1 and 2). Like most avian species, these seabirds can regulate breeding investment by adjusting egg number (terns and gulls) and size [29]. Egg volume in birds has an important genetic component [30], but in long-lived birds, its temporal variance constitutes a reliable indicator of environmental conditions (e.g. food availability) and individual breeding investment [31–33]. Moreover, egg volume can be correlated with chick growth and survival [34–37].

Using long-term monitoring data of Scopoli's shearwater, Sandwich tern, and Audouin's gull populations, we assess the influence of i) winter and spring values of the North Atlantic Oscillation index and ii) local-scale food-related variables measured in spring (during breeding) as predictors of the average egg volume in a clutch. For these local-scale variables, we included the factors potentially influencing food detectability and accessibility (foraging conditions and both intraspecific and interspecific competition) and per capita food abundance. Given the smaller area used by seabirds during the breeding period relative to the winter distribution, we expect local variables to be a better predictor than the North Atlantic Oscillation index on egg volume.

## Methods

### Field data and study area

Sandwich tern and Audouin's gull eggs were measured at the Ebro Delta while eggs of Scopoli's shearwaters were measured about 170km east, at Dragonera Island (Balearic archipelago; Fig 1). Previous studies based on observations and direct tracking of marked individuals indicate that adults of the three species forage actively within the Ebro Delta continental shelf [27, 28]; thus, we considered the continental shelf of a marine area of 100km radius centred on the

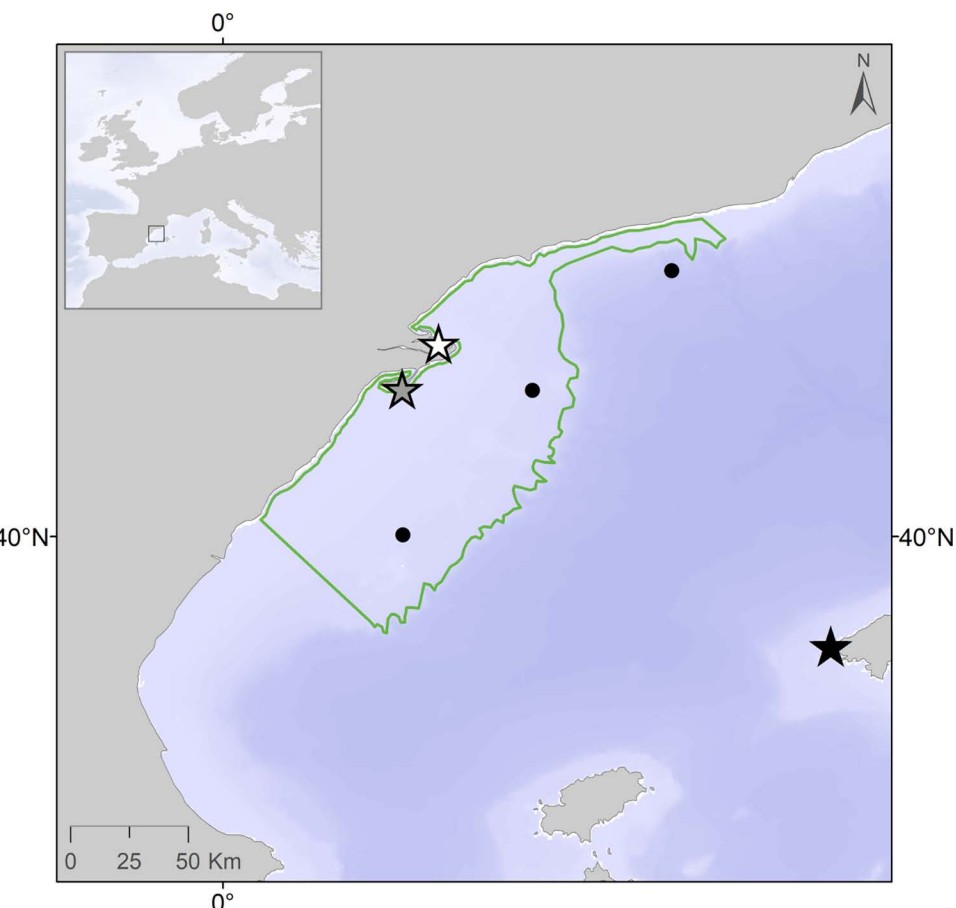

**Fig 1. Common foraging area (delimited by the green line) of studied populations during the breeding season within the Ebro Delta continental shelf (Western Mediterranean) and locations of breeding colonies of studied populations of the Scopoli's shearwater (black star), the Sandwich tern (grey star) and the Audouin's gull (white star).** Black dots represent stations where local climatic and oceanographic variables (wind speed and direction, wave height, and seawater turbidity) were measured to assess foraging conditions for studied populations during the early breeding season.

Ebro Delta as representative of their common foraging area (Fig 1). The three species mostly winter off the Atlantic coasts of Africa [27, 28], but gulls are partial migrants and part of their population remains along the Western Mediterranean coast [27]. For each species, we recorded the temporal variance of the annual mean volume of the modal clutch (hereafter, egg volume; N = 10573 clutches in total) as an indicator of the metabolic resources accumulated for breeding [31, 32]. Long-term data were collected for Scopoli's shearwaters (1440 one-egg clutches measured from 2001 to 2017), Sandwich terns (425 two-egg modal clutches measured between 2000 and 2016), and Audouin's gull (8708 three-egg modal clutches measured between 2001 and 2017). Eggs were measured with a digital caliper to ± 0.01mm and egg volume (V) was calculated in $cm^3$ according to the equation $V = K \times L \times W^2$ [38], where L = maximum egg length and W = maximum egg width and $K$ is a species-specific egg-shape constant: $0.510 \times 10^{-3}$ for shearwaters and terns and $0.467 \times 10^{-3}$ for gulls. Once measured, all the eggs were returned to the nest, therefore, Institutional Animal Care and Use Committee (IACUC) was not required. Despite being an important predictor of across individuals variation in egg volume [30], female size was not available for all nests and was assumed to be a random and additive component of the total variance within each colony [39]. The

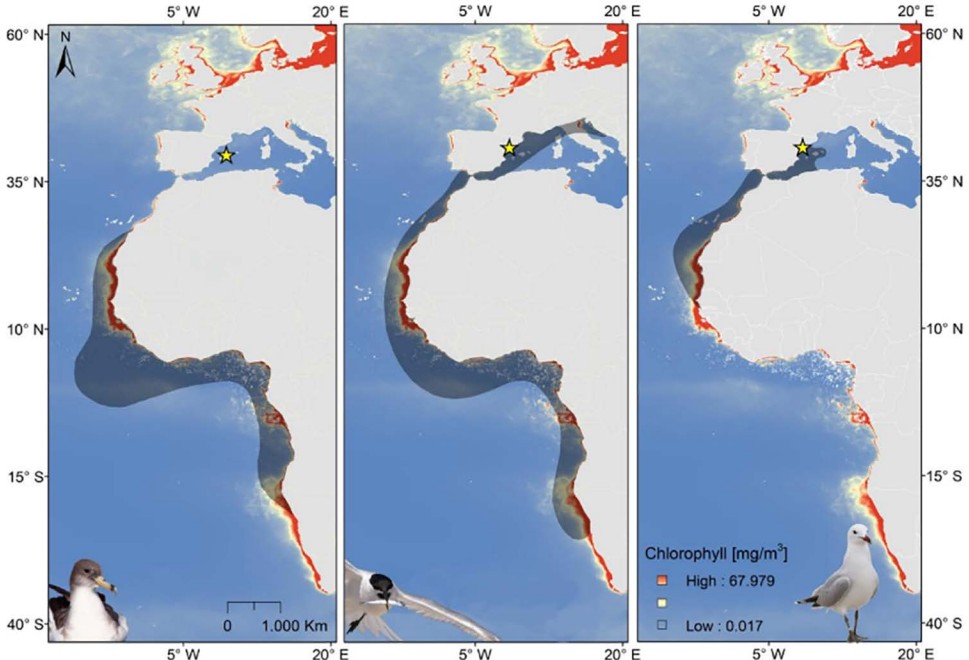

**Fig 2. Wintering areas (shaded areas) of studied populations of the Scopoli's shearwater (left; data from Reyes-Gonález et al 2017), the Sandwich tern (center; data from Institut Català d'Ornitologia), and the Audouin's gull (right; data from Bécares et al 2015).** Yellow stars indicate the location of the breeding colonies where eggs were measured. Mean annual sea surface concentration of chlorophyll-a for the period 2009–2013 is also shown (data obtained at http://data.unep-wcmc.org/datasets/37).

Governments of the Balearic Islands (Servei de Protecció d'Especies of the Conselleria de Medi Ambient of the Balearic Government) and Catalonia (Generalitat de Catalunya) provided the permits to work with each species studied. Access permits to protected areas were provided by Sa Dragonera Natural Park (Consell de Mallorca) and the Ebro Delta Natural Park respectively. The field studies did not involve endangered or protected species. All sampling procedures and/or experimental manipulations were reviewed and approved by the corresponding authorities after obtaining the field permit.

### Predictors of egg volume

**The large-scale climatic index.** We used winter means (December to March) of the station-based North Atlantic Oscillation index (wNAO) (https://climatedataguide.ucar.edu/climate-data/hurrell-north-atlantic-oscillation-nao-index-station-based) to assess the relative importance that winter conditions in the year $i$ (December $i$-1 to March $i$) has on predicting the egg volume of the following breeding season. Spring means (considering the species-specific early breeding season; see S4 Table) of the North Atlantic Oscillation (spNAO) were also used to assess the possible influence exerted by this climate index during spring months when the study species were laying eggs.

**Seawater turbidity, wave height, and wind speed and direction.** We assume that in the foraging process, animals first detect food and then try to access it. For this reason, we use the term 'accessibility' to refer to those physical barriers that hinder access to food once detected. To assess how local climatic and oceanographic variables influence the annual variance of the egg volume of the studied species we used monthly means of i) wind speed and direction, ii) wave height and iii) seawater turbidity as proxies of detectability and accessibility of food,

[40, 41]. Data on wind speed (m·s$^{-1}$), wind direction (degrees), and wave height (m) based on numeric modelling data were obtained from the SIMAR dataset at http://www.puertos.es/es-es/oceanografia/Paginas/portus.aspx. To account for a cumulative effect on foraging conditions, the number of days of winds blowing from each quartile (Q1: 0°-90°; Q2: 91°-180°; Q3: 181°-270° and Q4: 271° - 360°) were multiplied by the corresponding mean wind speed (days·m·s$^{-1}$). Finally, seawater turbidity was estimated by considering the diffuse attenuation coefficient of light at 490 nm (kd490) (1 Km$^2$ resolution) from multi-satellite observations (http://marine.copernicus.eu/services-portfolio/access-to-products/).

**Intra- and inter-specific competition during the early breeding season.** To assess the potential effect that intra-specific competition has on the temporal variance of the egg volume we used annual estimates of breeding pair numbers for each studied species. Population estimates of gulls and terns were obtained by direct counts. For Audouin's gull, estimates were based on annual censuses from three different colonies as these birds are known to share the same foraging area during the breeding season (Ebro Delta, Castellón, and Tarragona). For the Scopoli's shearwater, population size was estimated using the number of nests occupied in the study colony each year [42]. The effect of interspecific competition for food was assessed by considering the total number of breeding pairs of the three seabird species (S5 Table) as well as of the Yellow-legged gull *Larus michahellis*, an abundant generalist species competing for the same foraging resources (i.e. sardines, anchovies and trawling fishery discards) in the study area.

**Per capita food abundance during the early breeding season.** As a measure of natural and anthropogenic food resources, we used annual estimates of the *per capita* abundance of natural prey and fishery discards during the early period of the species-specific breeding season (S4 Table). Temporal variability in the abundance of natural prey was approximated by using the catch per unit effort (hereafter 'CPUE') of sardines *Sardina pilchardus* and anchovies *Engraulis encrasicolus*, the most abundant small pelagic species in the study area [43, 44] and prey of the three studied species [45–48]. CPUE was obtained by dividing the total landings (in Kg) of each species by the number of vessels of the main purse seine fleets in the area (S4 Table). Data on landings and number of fishing vessels were facilitated by the Direcció General de Pesca i Afers Marítims of the Generalitat de Catalunya. We used the sum of the main horsepower declared by trawl fleets in the study area (S4 Table) as a proxy of the abundance of fishery discards [33]. Trawl horsepower is a more precise estimate of discards generated than the number of vessels because the more horsepower the bigger are the nets used. Data on trawl horsepower were obtained from the European Commission Fleet Register at http://ec.europa.eu/fisheries/fleet/index.cfm. *Per capita* abundance of natural prey and fishery discards was calculated by dividing estimates of each resource type by the total number of breeding pairs of the main seabird species competing for each specific resource (S5 Table).

## Data analysis

We analyzed factors affecting the egg volume and their statistical interactions using generalized linear models in software R (R Development team 2014), with mean egg volume of the modal clutch (the most repeated clutch size among observations) as dependent variable. Covariates were centered and scaled to equalize their means and obtained comparable standard deviations. A diagnosis was made to check assumptions of the models (linearity normality, homogeneity of variance and independence of residuals). We began the analysis by calculating the correlation coefficient across all covariates described above to avoid the simultaneous presence of highly correlated covariates (collinearity) and only uncorrelated covariates (p value >0.05) were considered together in the models. For the best models, we also

run tests to check the variance inflation factors (VIFs). Only models were all VIF values were <3 were considered [49]. To check VIFs in models with interaction terms we first centered these covariates as suggested by [50]. All covariates were taken as fixed effects. Only model structures that made ecological sense according to species-specific diet and foraging strategies were compared. Information theory based on Akaike Information Criterion (AIC; [51, 52]) was used to select the best explanatory models. Best models were those with the lowest AIC and models with AIC differences ≤ 2 were considered equivalent [52]. The proportion of total annual variance in egg volume explained by covariates was calculated as [deviance (model constant)–deviance (model with covariate)] / [deviance (model constant)–deviance (model time-dependent)]. The resulting statistics can be used as an equivalent of the coefficient of determination, $R^2$ (hereafter $R^2$; see [53]).

## Results

### Scopoli's shearwater

For the Scopoli's shearwater, the model with the lowest AIC value included the additive effect of the wNAO and the statistical interaction between the wave height and the *per capita* abundance of fishery discards (Model 1 in Tables 1 and S1 and Fig 3A). According to this model, high values of wave height decreased the egg volume, even when fishery discards per capita were abundant. The wNAO had a negative effect on the egg volume whilst the effect of fishery discards was positive. When assessing wNAO and local conditions separately, the percentage of the total annual variance of the egg volume explained by the wNAO was 45% (Model 5 in Tables 1 and S1), while the model considering only local conditions (Model 3 in Tables 1 and S1) explained 66%. When tested simultaneously (wNAO, wave height, and fishery discards), these covariates explained 79% of the total annual variance of the egg volume (Model 1 in Table 1 and S1; Fig 3A). S1 Fig shows the relationship between the egg volume predicted by Model 1 and the egg volume observed. We did not find a significant effect of the number of potential competitors (Tables 1 and S1) nor of the spNAO.

**Table 1. Generalized linear models explaining egg volume variability (mean egg volume in a clutch) of the Scopoli's shearwater based on Akaike information criterion values (AIC) and Akaike weights (Wi).**

| Model | Notation | Deviance | df | AIC | ΔAIC | Wi |
|---|---|---|---|---|---|---|
| 1 | Winter NAO + WaveHeight * DiscardsPC | 40733.54 | 6 | 8911.61 | 0.00 | 0.85 |
| 2 | Winter NAO + WaveHeight + DiscardsPC | 40899.95 | 5 | 8915.48 | 3.87 | 0.12 |
| 3 | WaveHeight * DiscardsPC | 40966.80 | 5 | 8917.83 | 6.22 | 0.04 |
| 4 | Winter NAO + WaveHeight | 41035.65 | 4 | 8918.25 | 6.88 | 0.03 |
| 5 | Winter NAO | 41331.38 | 3 | 8926.59 | 14.98 | 0.00 |
| 6 | WaveHeight | 41478.83 | 3 | 8931.72 | 20.11 | 0.00 |
| 7 | Compet. by AG | 41558.13 | 3 | 8934.47 | 22.86 | 0.00 |
| 8 | Wind4Q | 41602.25 | 3 | 8936.00 | 24.39 | 0.00 |
| 9 | Discards PC | 41635.50 | 3 | 8937.15 | 25.54 | 0.00 |
| 10 | Wind3Q | 41693.72 | 3 | 8939.16 | 27.55 | 0.00 |

The best explanatory model (Model 1) is the one with the lowest AIC. In the notation: Winter NAO = winter North Atlantic Oscillation, Spring NAO = Spring North Atlantic Oscillation during the species-specific pre-laying period, SS = Scopoli's shearwater, YLG = Yellow-legged Gull, AG = Audouin's Gull, PC = per capita, Wind1Q, 2Q, 3Q, and 4Q = 1st, 2nd, 3rd and 4th quartile winds respectively (see methods section), Discards = fishery discards, Null model is an only-intercept model. Discards PC and Sardine PC consider the number of individuals of YLG+AG+SS.

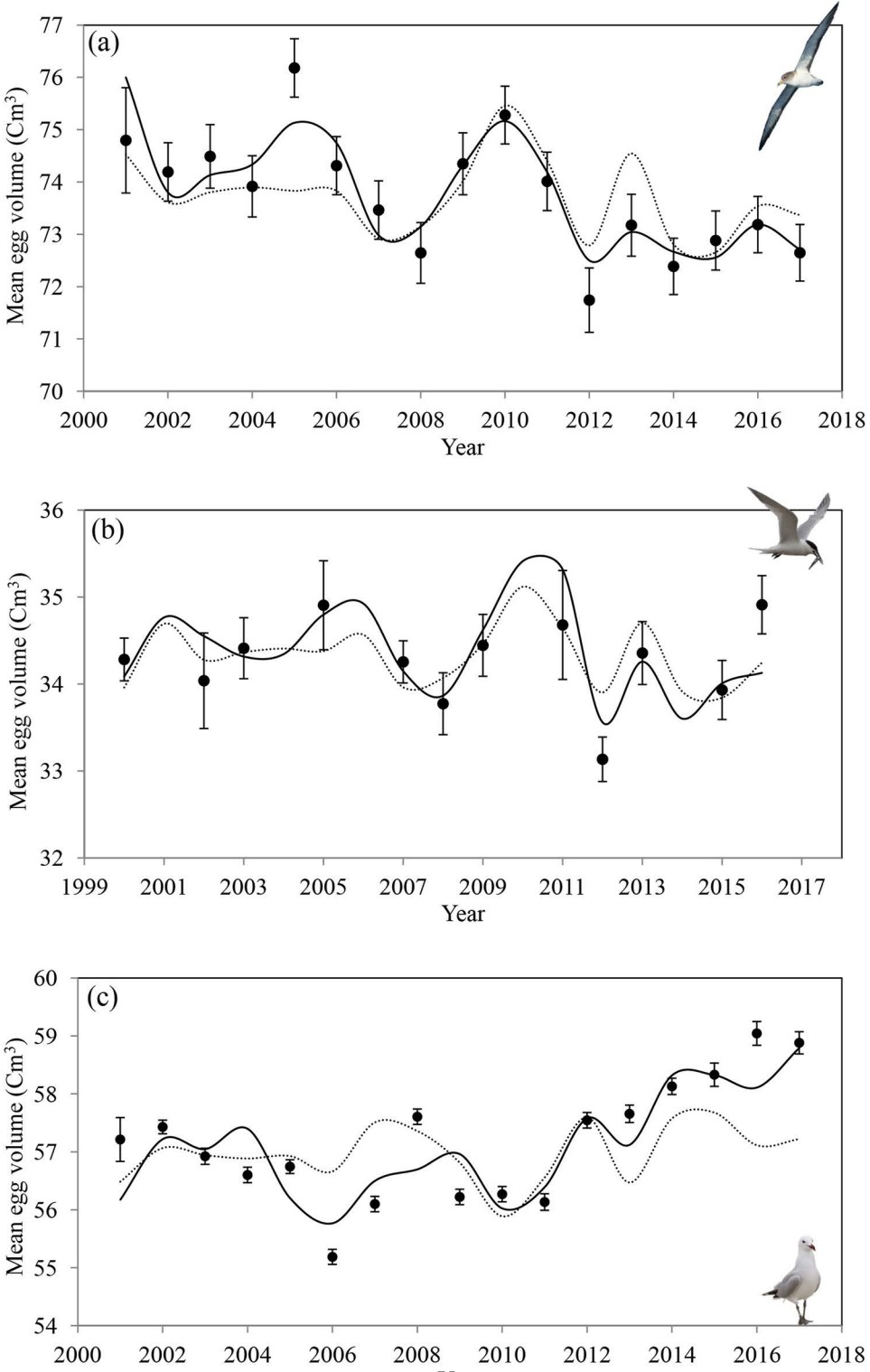

**Fig 3.** Time series with the observed mean egg volume ± standard error (black circles) and the expected mean egg volume according to best explanatory models (black line) in (a) the Scopoli's shearwater (Model 1, Tables 1 and S1), (b) the Sandwich tern (Model 1, Tables 2 and S2) and (c) the Audouin's gull (Model 1, Tables 3 and S3). Models only considering winter conditions (wNAO) are also shown (dotted line).

**Table 2. Generalized linear models explaining egg volume variability (mean egg volume in a clutch) of the Sandwich tern based on Akaike information criterion values (AIC) and Akaike weights (Wi).**

| Model | Notation | Deviance | df | AIC | ΔAIC | Wi |
|---|---|---|---|---|---|---|
| 1 | Winter NAO + Wind3Q + Turbidity | 1655.94 | 5 | 1787.71 | 0 | 0.23 |
| 2 | Winter NAO + Wind3Q | 1665.30 | 4 | 1788.10 | 0.39 | 0.19 |
| 3 | Winter NAO + Wind3Q + Turbidity + Compet. by YLG | 1650.57 | 6 | 1788.34 | 0.63 | 0.17 |
| 4 | Winter NAO + Wind3Q + Compet. by YLG | 1662.09 | 5 | 1789.28 | 1.57 | 0.10 |
| 5 | Winter NAO + Wind3Q + AnchovyPC | 1659.32 | 5 | 1789.58 | 1.87 | 0.08 |
| 6 | Winter NAO + Wind3Q * Turbidity | 1655.69 | 6 | 1789.65 | 1.94 | 0.09 |
| 7 | Winter NAO + Wind3Q + Wind1Q | 1664.41 | 5 | 1789.87 | 2.16 | 0.08 |
| 8 | Winter NAO | 1689.33 | 3 | 1792.16 | 4.45 | 0.02 |
| 9 | Wind3Q | 1690.22 | 3 | 1792.38 | 4.67 | 0.02 |
| 10 | Wind3Q + Turbidity | 1689.84 | 4 | 1794.30 | 6.59 | 0.01 |

The best explanatory model (Model 1) is the one with the lowest AIC. In the notation: Winter NAO = winter North Atlantic Oscillation, Spring NAO = Spring North Atlantic Oscillation during the species-specific pre-laying period, YLG = Yellow-legged Gull, PC = Per capita, Wind1Q, 2Q, 3Q, and 4Q = 1st, 2nd, 3rd and 4th quartile winds respectively (see methods section), Discards = fishery discards, Null model is an only-intercept model. Both, discards PC and anchovy PC consider the number of individuals of ST, YLG, AG, and SS.

## Sandwich tern

For the Sandwich tern, the best model suggested that egg volume was associated with the additive effect of the wNAO and food detectability and accessibility in the form of 3rd quartile winds and seawater turbidity (Model 1 in Tables 2 and S2 and Fig 3B). For all three covariates, positive values had a negative effect on the egg volume. Assessed alone, the wNAO explained 28% of the total annual variance (Model 8 in Tables 2 and S2), while local conditions explained 27% (Model 10 in Tables 2 and S2). All three covariates exerted a negative effect on the egg volume. When these covariates were tested simultaneously (Model 1 in Tables 1 and S1; Fig 3A), they explained 59% of the total annual variance. S2 Fig shows the relationship between the egg volume predicted by Model 1 and the egg volume observed. Models 2 to 6 including the additive effect of the interspecific competition exerted by Yellow-legged gulls, seawaters, turbidity, abundance of anchovy or the statistical interaction with the 3rd quartile winds, had similar explanatory power to Model 1 (i.e. ΔAIC values < 2; Tables 2 and S2). However, both the interaction and the additive terms in models 2 and 6 act as pretending variables. Pretending variables occur when after adding a new variable in a model, an ΔAIC~2 is obtained but, the deviance does not decrease [54] and should not be considered further (Appendix B in [55]).

## Audouin's gull

Model information theory indicated that Audouin's gull egg volume was influenced by the wNAO and the statistical interaction between the intra-specific and inter-specific competition (Model 1 in Tables 3 and S3 and Fig 3C). The wNAO had a positive effect on egg volume, while the effect of both intra- and inter-specific competition was negative. When tested individually, the percentage of the total annual variance of the egg volume explained by the wNAO was 24% (Model 10 in Tables 1 and S1), while the effect of competition (intra- and inter-specific) explained 59% (Model 4 in Tables 3 and S3). Tested simultaneously (Model 1 in Tables 3 and S3 and Fig 3C), these covariates explained 70% of the annual variance of the egg volume. S3 Fig shows the relationship between the egg volume predicted by Model 1 and the egg volume observed. Population density in this species was correlated with the abundance of natural prey, fishery discards, and inversely correlated with wave height (see S8 Table for more

**Table 3. Generalized linear models explaining egg volume variability (mean egg volume in a clutch) of the Audouin's gull based on Akaike information criterion values (AIC) and Akaike weights (Wi).**

| Model | Notation | Deviance | df | AIC | ΔAIC | W |
|-------|----------|----------|----|----|------|---|
| 1 | Winter NAO + Intrasp. compet. * Compet. by YLG | 94498.10 | 6 | 45482.82 | 0.00 | 1.00 |
| 2 | Winter NAO + Intrasp. compet. + Compet. by YLG | 94640.28 | 5 | 45493.91 | 11.09 | 0.00 |
| 3 | Winter NAO + Intrasp. compet. | 94695.19 | 4 | 45496.96 | 14.14 | 0.00 |
| 4 | Intraspecific competition * Competition by YLG | 95377.26 | 5 | 45561.45 | 78.63 | 0.00 |
| 5 | Intraspecific competition | 95560.62 | 3 | 45574.18 | 91.36 | 0.00 |
| 6 | Sardine PC | 95927.54 | 3 | 45607.54 | 124.72 | 0.00 |
| 7 | Wave height | 97782.89 | 3 | 45774.34 | 291.52 | 0.00 |
| 8 | Anchovy PC | 97811.19 | 3 | 45776.86 | 294.04 | 0.00 |
| 9 | Winter NAO | 98076.86 | 3 | 45800.48 | 317.66 | 0.00 |
| 10 | 4$^{th}$ q. winds | 98114.50 | 3 | 45803.82 | 321.00 | 0.00 |

The best explanatory model (Model 1) is the one with the lowest AIC. In the notation: Winter NAO = winter North Atlantic Oscillation, Spring NAO = Spring North Atlantic Oscillation during the species-specific pre-laying period, YLG = Yellow-legged Gull, AG = Audouin's Gull, SS = Scopoli's shearwater, PC = Per capita, Discards = fishery discards, Null model is an only-intercept model. Discards PC, Sardine PC, and Anchovy PC consider the total number of individuals of the three study species.

details). Therefore, to avoid collinearity these covariates were not considered together in our models. Contrary to the other species considered, the effect of the wNAO on the egg volume of the Audouin's gull was positive.

## Discussion

The effects of the wNAO and the local variables measured in spring were different for each study species, likely due to the differences in life histories, foraging strategies, and wintering geographical ranges. However, for all three species, the temporal variation in average egg volume was better explained when both (local variables measured in spring and wNAO) were considered together.

The additive effect of wNAO on breeding investment found here would indicate a *carry-over* effect of winter conditions on subsequent reproductive seasons [56, 57]. This effect was especially important for shearwaters showing the slowest life history strategy of the three study species [58]. This result also suggests that Cory's Shearwaters act as capital breeders, however further research is needed in this regard, so this possibility should be taken with caution. The wNAO and winter anomalies influence wind speed and direction and the wave height in the Atlantic Ocean and the Mediterranean Sea [59]. High values of wNAO likely drove adverse foraging conditions for shearwaters during wintering and a poorer body condition for the subsequent breeding season. The wNAO influenced negatively the egg volume of shearwaters and sandwich terns but had a positive influence on the egg volume of the Audouin's gull. A plausible explanation is that many gulls remain in the Western Mediterranean during winter (see Fig 2; [27, 28]), where positive values of the wNAO are associated with less stormy winters (see e.g. [60]). [8] found that bird species wintering in the Mediterranean area had different responses to the wNAO when compared to species wintering in northern Europe. Another non-exclusive explanation is that intra- and inter-specific competition has greater importance than other effects especially for Audouin's gulls, which are outcompeted by the larger yellow-legged gull for the same size and type of food [61]. Different responses to the wNAO could also be associated with differences in species-specific foraging strategies (e.g. terns and gulls can cope with adverse weather conditions by feeding in sheltered coastal areas, while shearwaters only feed in the open sea). Finally, these differences could also be related to the fact that the study species have different evolutionary life histories: terns and gulls are

multiparous, while shearwaters lay a single egg. Interestingly, local variables related to food detectability and accessibility (oceanographic physical features; see e.g. [40, 41, 62]) and competition during the early breeding season were more important than food abundance. This implies that food abundance *per se* does not necessarily translate into food intake for predators and that the role played by detectability and accessibility of food in ecological processes deserves more consideration [21, 63, 64].

The relative importance of the variables explaining food detectability and accessibility (wave height, wind speed and direction, seawater turbidity, and competition) also changed for each study species, and once again, this is likely due to differences in their foraging strategies. Larger waves may drive natural prey to deeper waters affecting their detectability and accessibility to shearwaters [40]. Wind may act in opposite ways on fishing conditions depending on its intensity [65]. Strong winds may negatively affect terns' flying trajectories when they pounce on their prey during fishing influencing both, detectability and accessibility of food [40, 66], but favourable winds can result in important energy savings by seabirds on displacements (see e.g. [67–69]). Although weak, we also detected a negative effect of water turbidity on the egg volume of Sandwich terns. Previous studies have shown that water turbidity negatively influences prey detectability of Sandwich terns [41] and other seabird species (see e.g. [20, 70]).

A negative effect of intra- and interspecific competition (exerted by the Yellow-legged gull) was retained for the Audouin's gull only. Competition between the two species of gulls has been previously reported in the Western Mediterranean, and we found that as intraspecific competition increases, interspecific competition decreases [61, 71]. The important role of density-dependence for this species is not surprising, considering that the study colony hold up to 73% of the total world population [45, 72]. Inter- and intra-specific competition for food seemed to overcome the influence of climatic and local oceanographic variables (see [73]). However, the effect of density-dependence was not retained for terns and shearwaters, whose densities were likely underestimated by missing birds coming from neighbouring colonies.

## Conclusions

Our results provide new insights on the relative influence of large-scale climatic indices *vs* local variables on egg volume as a proxy of breeding investment. We showed that both large-scale climatic index and local variables are correlated with breeding investment because they operate at different spatio-temporal scales. The wNAO index acts in the form of a *carry-over* effect arising from winter conditions, while local conditions act as proximate causes of food intake. Finally, and in contrast to what is commonly assumed, food abundance does not necessarily translate into individual food intake. Large-scale climate indices present several advantages as indicators of regulating forces of ecosystems [12, 74], especially when animals are widely distributed in space (e.g. across wintering regions), but local variables may be more important and can provide a better explanation of processes affecting conditions for foraging, including food detectability and accessibility. Further research should focus on fine-tuning the mechanisms through which local variables affect food intake, for example by coupling foraging activity with tracking data and sea-state variables.

## Supporting information

**S1 Table. Models explaining the egg volume variability of the Scopoli's shearwater.** (DOCX)

**S2 Table. Models explaining the egg volume variability of Sandwich terns.**
(DOCX)

**S3 Table. Models explaining the egg volume variability of the Audouin's gull.**
(DOCX)

**S4 Table. Species-specific early breeding periods of studied populations.**
(DOCX)

**S5 Table. Foraging interactions and competition for food among the study species.**
(DOCX)

**S6 Table. Correlation matrix for the Scopoli's shearwater.**
(DOCX)

**S7 Table. Correlation matrix for Sandwich terns.**
(DOCX)

**S8 Table. Correlation matrix for the Audouin's gull.**
(DOCX)

**S9 Table. Annual values of egg volume and covariates for the Scopoli's shearwater.**
(DOCX)

**S10 Table. Annual values of egg volume and covariates for Sandwich terns.**
(DOCX)

**S11 Table. Annual values of egg volume and covariates for the Audouin's gull.**
(DOCX)

**S12 Table. Estimates ± SE of the best model for the Scopoli's shearwater.**
(DOCX)

**S13 Table. Estimates ± SE of the best models for Sandwich terns.**
(DOCX)

**S14 Table. Estimates ± SE of the best model for the Audouin's gull.**
(DOCX)

**S1 Fig. Egg volume predicted v.s. observed for the Scopoli's shearwater.**
(DOCX)

**S2 Fig. Egg volume predicted v.s. observed for Sandwich terns.**
(DOCX)

**S3 Fig. Egg volume predicted v.s. observed for the Audouin's gull.**
(DOCX)

**S1 File. Supporting information references.**
(DOCX)

## Acknowledgments

We are grateful to everyone who helped with fieldwork and provided data, images, and logistic support, especially the staff of the Ebro Delta Natural Park.

## Author contributions

**Conceptualization:** Giacomo Tavecchia.

**Data curation:** Enric Real, Daniel Oro, Albert Bertolero, José Manuel Igual, Ana Sanz-Aguilar, Meritxell Genovart, Giacomo Tavecchia.

**Formal analysis:** Enric Real.

**Funding acquisition:** Daniel Oro, Giacomo Tavecchia.

**Investigation:** Enric Real.

**Methodology:** Enric Real, Manuel Hidalgo, Giacomo Tavecchia.

**Project administration:** Daniel Oro, Giacomo Tavecchia.

**Resources:** Giacomo Tavecchia.

**Supervision:** Daniel Oro, José Manuel Igual, Giacomo Tavecchia.

**Validation:** Giacomo Tavecchia.

**Writing – original draft:** Enric Real.

**Writing – review & editing:** Daniel Oro, Albert Bertolero, José Manuel Igual, Ana Sanz-Aguilar, Meritxell Genovart, Manuel Hidalgo, Giacomo Tavecchia.

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
