## [Decision Letter · Decision Letter 0]

26 Oct 2021

PONE-D-21-28485It’s not all availability: detectability and accessibility to food also explain breeding investment in long-lived animalPLOS ONE

Dear Dr. Real,

Thank you for submitting your manuscript to PLOS ONE. After careful consideration, we feel that it has merit but does not fully meet PLOS ONE’s publication criteria as it currently stands. Therefore, we invite you to submit a revised version of the manuscript that addresses the points raised during the review process.

We look forward to receiving your revised manuscript.

Kind regards,

Vitor Hugo Rodrigues Paiva, Ph.D.

Academic Editor

PLOS ONE

Journal Requirements:

“We are grateful to everyone who helped with fieldwork and provided data, images, and logistic support, especially the staff of the Ebro Delta Natural Park. Funds were supplied by grants CGL2013-42203-R and CGL2017-85210-P (MCIU/AEI/FEDER, UE). The study also received funding from the European Commission’s Horizon 2020 Research and Innovation Program under Grant Agreement no. 634495 for the project Sci¬ence, Technology, and Society Initiative to Minimize Unwanted Catches in European Fisheries (MINOUW). MG was partially funded by Govern Balear. ASA was supported by a Ramón y Cajal contract (RYC-2017- 22796).”

We note that you have provided additional information within the Acknowledgements Section. Please note that funding information should not appear in the Acknowledgments section or other areas of your manuscript. We will only publish funding information present in the Funding Statement section of the online submission form.

“Funds were supplied by grants CGL2013-42203-R and CGL2017-85210-P (MCIU/AEI/FEDER, UE) respectively. The study also received funding from the European Commission’s Horizon 2020 Research and Innovation Program under Grant Agreement no. 634495 for the project Sci-ence, Technology, and Society Initiative to Minimize Unwanted Catches in European Fisheries (MINOUW). MG was partially funded by Govern Balear. ASA was supported by a Ramón y Cajal contract (RYC-2017- 22796).”

 “Funds were supplied by grants CGL2013-42203-R and CGL2017-85210-P (MCIU/AEI/FEDER, UE) respectively. The study also received funding from the European Commission’s Horizon 2020 Research and Innovation Program under Grant Agreement no. 634495 for the project Sci-ence, Technology, and Society Initiative to Minimize Unwanted Catches in European Fisheries (MINOUW). MG was partially funded by Govern Balear. ASA was supported by a Ramón y Cajal contract (RYC-2017- 22796).”

7. We note that you have stated that you will provide repository information for your data at acceptance. Should your manuscript be accepted for publication, we will hold it until you provide the relevant accession numbers or DOIs necessary to access your data. If you wish to make changes to your Data Availability statement, please describe these changes in your cover letter and we will update your Data Availability statement to reflect the information you provide.

Reviewers' comments:

Reviewer's Responses to Questions

**Comments to the Author**

1. Is the manuscript technically sound, and do the data support the conclusions?

Reviewer #1: Yes

Reviewer #2: Yes

Reviewer #3: Partly

2. Has the statistical analysis been performed appropriately and rigorously? 

Reviewer #1: Yes

Reviewer #2: Yes

Reviewer #3: No

3. Have the authors made all data underlying the findings in their manuscript fully available?

Reviewer #1: No

Reviewer #2: Yes

Reviewer #3: Yes

4. Is the manuscript presented in an intelligible fashion and written in standard English?

Reviewer #1: Yes

Reviewer #2: Yes

Reviewer #3: Yes

5. Review Comments to the Author

Reviewer #1: This paper correlates long-term monitoring data on egg volume in three model marine bird species and values of the NAO Index as well as local-scale food related variables. The manuscript is well written and provides an interesting contribution to this topic. I have several comments to further improve clarity of the paper.

Keywords should be listed alphabetically.

The Introduction section would benefit from adding some hypotheses and/or predictions.

L 99 and elsewhere: Please avoid ") (".

L 176: Please introduce and justify the use of interactions.

L 178: Please remove dash in "R- Development team 2014".

L 179: Or simply by 1000?

L 191: The study species is in singular, but in L 205 and 219 plural.

L 220-2: Please specify the directions of the significant relationships.

L 271-3: Not fully clear to me.

L 299: Here I would write about conditions rather than variables.

L 585: Please better explain wind quartiles or refer to the methods section (see also supplement).

Fig. 3: Please improve figure resolution.

Supplement: Please use decimal points and the same number of decimals in tables.

Table S7: P-values cannot be higher than 1. Do the authors mean <0.05?

Reviewer #2: It’s not all availability: detectability and accessibility to food also explain breeding investment in long-lived animals

General comments

This paper addresses some important and largely ignored issues in ecology, specifically the relative roles of large-scale and local-scale drivers, and the metrics used to explain (eg food abundance vs per capita abundance), demographic response variables for long-lived species. The study used a comprehensive data set from three species with differing life history strategies. I found the paper to be well-written, concise and with a clearly defined analysis and message. I have no major comments or criticisms, but list some minor comments below. I recommend the paper publication is suitable for publication in PLOS ONE

Specific comments

Line 73.’ Bet-hedgers species’ should be ‘bet-hedger species’ or ‘bet-hedging species’?

Line 74. I’m not sure what you mean by ‘slower’ life history. Can you expand a little and/or contrast more explicitly with a bet-hedging strategy?

Lines 89-91. You mention two issues (food detectability and availability) and cite three related processes (sea conditions, per capita food abundance and competition) without making clear their associations. Presumably sea conditions relates to detectability and per capita abundance and competition to availability? I think it would help the reader to make the explicit association here between the issues and processes

Lines 102-103. In reading this description of the common foraging area I was expecting to see a semi-circle of 100 km radius centred on the Ebro delta in Fig 1, but the green delineation seems to be indicated by a combination of the shelf area 100 km to either side of the delta along the coast. You could perhaps fine-tune the description a bit.

Lines 131-132. The issues of detectability and availability are central to the paper, but its not clear to me (yet) how they differ in principle. For example, if it is hard to detect food because the water is turbid, then it could be argued that the food is also inaccessible even if it is abundant, because it is unseen by whatever process. I am most familiar with these terms in the abundance estimation literature, where say in an aerial survey of a marine species, availability refers to whether an animal is near the surface and therefore potentially detectable, and detectability refers to the chance of seeing it given it is near the surface and therefore available for detection. Can you define these terms in the context of your paper and the three species which have differing foraging strategies? I see on lines 270-271 the example of prey being at greater depths is seen to influence both detectability and availability as if they are the same thing? Is this the intent?

Line 173. Insert ‘by’ between calculated and divided: ie ‘calculated by dividing’

Lines 183-184. There is an imbalance of parentheses here

Lines 213-214. Interspecific competition of sandwich terns with yellow legged gulls…is this consistent with their dive depths (terns are short diving plungers)

Lines 282-283. I am interested in whether the finding of inter-specific competition between Audouins gull and the yellow legged gull is related to them having similar foraging strategies in relation to depth. You say early in the paper that that gulls are surface feeders, and if this applied to both these species, you would expect a higher chance of interspecific competition between these two species than with the other species, and this could be useful to have in the discussion.

Reviewer #3: The authors present an analysis of annual egg-volume in three seabird species. Overall, I think there is merit in this analysis, but the authors need to give much more detail in their methods and results. The author frame the work in terms of comparing large-scale and local climate variables, but I am not convinced their methods properly compared these two variables. They considered both, but there are no details on statistically evaluating whether one had a smaller or larger impact than the other. I’d also like to see the methods section written so that another research could use the same dataset and replicate the analysis exactly. I am also wary of the conclusions drawn in the discussion of the study. The study uses many proxies of different climate variables to identify statistical correlations. The authors need to be more cautious in their interpretation to not assume they are observing actual effects of these variables on egg-volume. Finally, I’d like to see the estimates from each of the top models identified as well as some plots showing the relationship between predictors and egg-volume. I think it is fine if these go in the supplementary, but they are needed to properly evaluate the conclusions drawn in this study.

Line 79-83: I think the wording here is not careful enough. It is the among individual variation that matters more for your argument here. If it is of environmental origin, but has a permanent impact of egg size throughout the lifetime of an adult it will make egg-size just as inflexible in response to environmental conditions. However, heritabilities and repeatabilities are dependent on the environment they are measured in and don’t tell use much about flexibility in different environments– so the measurements in Christians 2002 don’t have much bearing on whether or not egg size will be flexible or plastic in response to environmental conditions. I think you should just remove the first part of the sentence and remark that egg-size is used as an indicator of environmental conditions.

Line 83-85: Egg volume can be correlated with chick growth and survival. I think it is important to not suggest that it is always correlated with chick growth and survival

Line 86: “Using long-term monitoring of data of these three species...” Clearly name species here. You’ve mentioned them above, but it isn’t clear that they are the focus of your paper.

Missing in the introduction is a brief sentence indicating why the different variables to affect egg volume. Is this through food resources, stressful conditions?

Are egg measurements repeated across females? Do you have identification of the females for each clutch? Observations are likely not independent and you have not accounted for this non-independence. Even if you do not have size measurements it might be important to consider female ID. If this is a smaller subset of the data it would be valuable to see the analysis with female ID or female size and see how these results compare to your overall results.

Data analysis

Do you only assess linear trends in your analysis? How did you decide to include and evaluate interactions?

Line 131-135: You arent really assessing how these variables affect detectability and accessibility of food. You are assuming that these proxies you use are using are related to the detectability and accessibility of food for foraging birds. You make the assumption that a relationship between and egg-volume is because of a relationship with food. Please reword.

Line 133: Table S4 does not seem to have any relationship to this sentence?

You use temporal variance of egg volume throughout, but I think it might be clearer to use among annual variance in egg volume. Temporal could be variance within a breeding season.

In the intra and inter specific competition section please clarify how you evaluated the effects of your population counts on each species. Were the count values included as fixed effects in your models?

Line 180-182: What did you do about collinearity? You need to explain more here.

Line 183-184: More information is needed here. How was model selection conducted. How did you conduct model selection? Did you compare all possible model structures? Did you try interactions? Did you use the dredge function in MuMIn? Please explain your process.

Results

It will be important to see the data behind the relationships suggested in your top models. I want to see the relationship between egg-volume and and your predictors.

For example, what does the egg-volume look like for eggs measured when wave height is high and discards are high vs low wave height and low discards vs low wave height and high discards?

You need to report your actual model results somewhere. The top model for each species will be fine, but it is needed to evaluate your conclusions!

Line 216-218: You need to explain what “pretending variables” are.

Discussion

Discussion would be clearer if separated by species or by large versus small scale effects. Headings would also be helpful if allowed in journal format.

Line 232 -236: If for all three species egg- volume was better predicted by including both local and large-scale variables, how can you say the relative importance was different? How did you evaluate this? Is the magnitude of the effect sizes different? If you mean that the effects of the same variable were different among species than you need to say this instead.

Line 239-241: Please explain the importance of this effect with respect to the slow life-history of the shearwaters. Why would we expect a carry-over effect in a species with a slow-life history? Especially with respect to reproductive investment. Are shearwaters income or capital breeders?

Line 258-259: Please explain the logic linking this sentence to the topic of this paragraph?

Line 263: Is it possible your variables are a poor proxy for food abundance?

Line 270: Can you provide better support for this possibility? What are the main prey sources of the shearwaters and is there evidence they are affect by larger waves?

Line 277: Reference software error on this line?

Line 291-293: Other possible explanations here?

Conclusions

My main feedback for the conclusions is to change the language so it is less certain. For example replace uses of the word influence with correlated. I think it is important to remember all the results are correlations and do not necessarily concretely prove the conclusions described here.

Figure caption three is confusing. Are these predictions from the best models for each species or are these the predictions from models with both spring and winter conditions? Please clarify.

6. PLOS authors have the option to publish the peer review history of their article (what does this mean?). If published, this will include your full peer review and any attached files.

Reviewer #1: No

Reviewer #2: No

Reviewer #3: No

---

## [Author Response · Author response to Decision Letter 0]

19 Feb 2022

Reviewer #1: This paper correlates long-term monitoring data on egg volume in three model marine bird species and values of the NAO Index as well as local-scale food related variables. The manuscript is well written and provides an interesting contribution to this topic. I have several comments to further improve clarity of the paper.

Keywords should be listed alphabetically.

- Thank you, we have modified this and keywords are now listed alphabetically

The Introduction section would benefit from adding some hypotheses and/or predictions.

- Yes, our hypotheses is in the last sentence of the Introduction section “Given the smaller area used by seabirds during the breeding period relative to the winter distribution, we expect local variables to be a better predictor than the North Atlantic Oscillation index on egg volume”.

L 99 and elsewhere: Please avoid ") (".

- Thank you, this has been already corrected.

L 176: Please introduce and justify the use of interactions.

- Totally agree, we have added a sentence which introduce and justify the evaluation of potential statistical interactions between covariates.

L 178: Please remove dash in "R- Development team 2014".

- Thank you, dash in "R- Development team 2014" has been removed.

L 179: Or simply by 1000?

- We agree that this aids in better readability, so it has been modified as well.

L 191: The study species is in singular, but in L 205 and 219 plural.

- Thank you, now they are all in singular.

L 220-2: Please specify the directions of the significant relationships.

- Yes, this information had been specified for the Scopoli’s shearwaters only. Now we have incorporated this information not only for Audouin gulls, but also for Sandwich terns.

L 271-3: Not fully clear to me.

- Yes, the sentence was not much clear. Now we have simplified this sentence for better understanding.

L 299: Here I would write about conditions rather than variables.

- Thank you, we have changed variables by conditions.

L 585: Please better explain wind quartiles or refer to the methods section (see also supplement).

- Thank you. Now we have added “(see methods section)” in Tables 1,2, S1, S2 and S3.

Fig. 3: Please improve figure resolution.

- Figures were submitted in TIF format and high resolution. However figures have no high resolution in the final draft in pdf.

Supplement: Please use decimal points and the same number of decimals in tables.

- Thank you, now we have used decimal points in the Supplementary material and al the tables have the same number of decimals.

Table S7: P-values cannot be higher than 1. Do the authors mean <0.05?

- Sorry for this mistake, as you say what we mean is p value <0.05. We have corrected in Tables S6. S7 and S8.

Reviewer #2: It’s not all availability: detectability and accessibility to food also explain breeding investment in long-lived animals

General comments

This paper addresses some important and largely ignored issues in ecology, specifically the relative roles of large-scale and local-scale drivers, and the metrics used to explain (eg food abundance vs per capita abundance), demographic response variables for long-lived species. The study used a comprehensive data set from three species with differing life history strategies. I found the paper to be well-written, concise and with a clearly defined analysis and message. I have no major comments or criticisms, but list some minor comments below. I recommend the paper publication is suitable for publication in PLOS ONE

Specific comments

Line 73.’ Bet-hedgers species’ should be ‘bet-hedger species’ or ‘bet-hedging species’?

- Thank you, we have changed “bet-hedger species” by “bet-hedgers”.

Line 74. I’m not sure what you mean by ‘slower’ life history. Can you expand a little and/or contrast more explicitly with a bet-hedging strategy?

- We agree this sentence was not clear enough and maybe unnacurate. Mow we have simplified this sentence as follows: “Furthermore, terns and gulls can modulate clutch size depending on environmental conditions, whereas shearwaters lay a single egg.”

Lines 89-91. You mention two issues (food detectability and availability) and cite three related processes (sea conditions, per capita food abundance and competition) without making clear their associations. Presumably sea conditions relates to detectability and per capita abundance and competition to availability? I think it would help the reader to make the explicit association here between the issues and processes.

- The issues we mention here are detectability and accessibility of food but not its availability. Sea conditions influence both, the detectability of food (eg turbidity, waves…) and its accessibility. For instance, in high swell conditions, natural prey are found at greater depths being less accessible. Competition also influence food accessibility. For example, intra- and –interspecific competition during the discard process make discards less accessible. However we agree this sentence should be improved. We have changed this sentence as follows: “For these local-scale variables, we included the factors influencing food detectability and accessibility (foraging conditions and both intraspecific and interspecific competition) and per capita food abundance.”

Lines 102-103. In reading this description of the common foraging area I was expecting to see a semi-circle of 100 km radius centred on the Ebro delta in Fig 1, but the green delineation seems to be indicated by a combination of the shelf area 100 km to either side of the delta along the coast. You could perhaps fine-tune the description a bit.

- Thank you, we have also changed this sentence as follows: “thus, we considered the continental shelf of a marine area of 100km radius centred on the Ebro Delta as representative of their common foraging area”

Lines 131-132. The issues of detectability and availability are central to the paper, but its not clear to me (yet) how they differ in principle. For example, if it is hard to detect food because the water is turbid, then it could be argued that the food is also inaccessible even if it is abundant, because it is unseen by whatever process. I am most familiar with these terms in the abundance estimation literature, where say in an aerial survey of a marine species, availability refers to whether an animal is near the surface and therefore potentially detectable, and detectability refers to the chance of seeing it given it is near. Can you define these terms in the context of your paper and the three species which have differing foraging strategies?

- Thanks for this observation. We have added the following sentence in the Methods section to clarify this: “We assume that in the foraging process, animals first detect food and then try to access it. Therefore, we use the term accessibility to refer to those physical barriers making the food item difficult to reach once it is detected.”

At the end of the Introduction section, in this sentence “For these local-scale variables, we included the factors potentially influencing food detectability and accessibility”, we have added the term “potentially”, because we hipothesize that these factors may influence detectability and accessibility of food and ultimately egg volume.

Regarding the relationship between these terms (detectability and accessibility) and species-specific feeding strategies, we have improved the explanations of feasible mechanisms in the Discussion section, specifying whether each covariate considered affects detectability, accessibility, or both, especially in those sentences where it had not been made clear enough..

I see on lines 270-271 the example of prey being at greater depths is seen to influence both detectability and availability as if they are the same thing? Is this the intent?

- No, this was not the intention. What we are referring to here is that when prey are located deeper: i) they will be more difficult to detect from the air and ii) they will be less accessible because deeper and longer dives will be required by seabirds. This is also well illustrated in [39 and 63].

Line 173. Insert ‘by’ between calculated and divided: ie ‘calculated by dividing’

- Thanks. It has been already inserted.

Lines 183-184. There is an imbalance of parentheses here

- Thanks, this has been already corrected too.

Lines 213-214. Interspecific competition of sandwich terns with yellow legged gulls…is this consistent with their dive depths (terns are short diving plungers)

- We consider that competition could occur in at least two different ways here. Firstly, when both species compete for discards and secondly, due to the kleptoparasitism exerted by yellow-legged gulls on terns, which probably affects both discards and natural prey.

Lines 282-283. I am interested in whether the finding of inter-specific competition between Audouins gull and the yellow legged gull is related to them having similar foraging strategies in relation to depth. You say early in the paper that that gulls are surface feeders, and if this applied to both these species, you would expect a higher chance of interspecific competition between these two species than with the other species, and this could be useful to have in the discussion.

- Yes, competition between these two species has been well documented by several authors, especially for fishery discards. We have modified a sentence at the end of the Discussion section which includes some references in this regard. Some studies have also shown that in the western Mediterranean, yellow-legged gulls and Audouin's gulls compete for discards in the vicinity of fishing boats. In contrast, shearwaters (much better divers) often position themselves 30-50m to the rear of boats to reach discards that have sunk deeper.

Reviewer #3: The authors present an analysis of annual egg-volume in three seabird species. Overall, I think there is merit in this analysis, but the authors need to give much more detail in their methods and results. The author frame the work in terms of comparing large-scale and local climate variables, but I am not convinced their methods properly compared these two variables. They considered both, but there are no details on statistically evaluating whether one had a smaller or larger impact than the other.

- Thanks for the comment. In one hand, we also include models where covariates were taken one by one to assess their relative importance in terms of AIC. These AICs are found in the supporting information (Tables S1, S2 and S3). On the other hand, we assessed the relative importance of some covariates by calculating their R2 (see Data Analysis in the Methods section for more details).

I’d also like to see the methods section written so that another research could use the same dataset and replicate the analysis exactly.

- Yes, now datasets are available as they have been uploaded to a public repository. The links to access to datasets have been also delivered to Plos One. In the methods section there is all necessary information to replicate the analysis exactly.

I am also wary of the conclusions drawn in the discussion of the study. The study uses many proxies of different climate variables to identify statistical correlations. The authors need to be more cautious in their interpretation to not assume they are observing actual effects of these variables on egg-volume. Finally, I’d like to see the estimates from each of the top models identified

- Thank you for the comment. These estimates are now in the supporting information in Tables S12, S13 and S14.

as well as some plots showing the relationship between predictors and egg-volume. I think it is fine if these go in the supplementary, but they are needed to properly evaluate the conclusions drawn in this study.

- Thanks again, these plots are now in the supporting information (Figures S1, S2 and S3).

Line 79-83: I think the wording here is not careful enough. It is the among individual variation that matters more for your argument here. If it is of environmental origin, but has a permanent impact of egg size throughout the lifetime of an adult it will make egg-size just as inflexible in response to environmental conditions. However, heritabilities and repeatabilities are dependent on the environment they are measured in and don’t tell use much about flexibility in different environments– so the measurements in Christians 2002 don’t have much bearing on whether or not egg size will be flexible or plastic in response to environmental conditions. I think you should just remove the first part of the sentence and remark that egg-size is used as an indicator of environmental conditions.

- Several researchers have said that the egg volume is not a good indicator of environmental conditions since this parameter is strongly dependent on females' size. We are fully aware of the importance of female size, which in some birds can account for up to 80% of the total egg volume, but many studies show that the other ~20% is much more flexible and dependent on environmental conditions, including food availability. We consider it is very important to clarify this so, that's why we decided to include this sentence.

Line 83-85: Egg volume can be correlated with chick growth and survival. I think it is important to not suggest that it is always correlated with chick growth and survival

- Thank you. We have already changed this.

Line 86: “Using long-term monitoring of data of these three species...” Clearly name species here. You’ve mentioned them above, but it isn’t clear that they are the focus of your paper.

- Thanks for the comment. We have modified this sentence including the names of each study species.

Missing in the introduction is a brief sentence indicating why the different variables to affect egg volume. Is this through food resources, stressful conditions?

- In Lines 86-94 in the Introduction section we explain that we use different variables potentially affecting food detectability and accessibility and per-capita food abundance (all of which are food-related variables that affect food intake) as predictors of the egg volume. We consider that it can be assumed by the reader that these variables influence egg volume through food intake. However, we can change this paragraph if necessary.

Are egg measurements repeated across females? Do you have identification of the females for each clutch? Observations are likely not independent and you have not accounted for this non-independence. Even if you do not have size measurements it might be important to consider female ID. If this is a smaller subset of the data it would be valuable to see the analysis with female ID or female size and see how these results compare to your overall results.

- We agree that considering this would be valuable but, females data was not available so we assumed to be a random and additive component of the total variance.

Data analysis

Do you only assess linear trends in your analysis?

- Yes, taking into account the number of covariates and spatial scales, we considered that GLMs are an appropriate technique in this case.

How did you decide to include and evaluate interactions?

- When considering possible scenarios in the foraging process. For example, we suspect that when sea conditions are very bad, it is more complicated for seabirds to reach discarded items so we decided to evaluate this interaction.

Line 131-135: You arent really assessing how these variables affect detectability and accessibility of food. You are assuming that these proxies you use are using are related to the detectability and accessibility of food for foraging birds. You make the assumption that a relationship between and egg-volume is because of a relationship with food. Please reword.

- Thank you. We have reworded these paragraph to clarify that these variables are used as proxies of food detectability and accessibility.

Line 133: Table S4 does not seem to have any relationship to this sentence?

- We apologize, this was a mistake. We have removed the words "Table S4" from this part of the manuscript.

You use temporal variance of egg volume throughout, but I think it might be clearer to use among annual variance in egg volume. Temporal could be variance within a breeding season.

- Thanks again, we agree and we have changed ‘temporal variance’ by ‘annual variance’ through the text except on those cases were we specify that it is annually.

In the intra and inter specific competition section please clarify how you evaluated the effects of your population counts on each species. Were the count values included as fixed effects in your models?

- In this study we have not included random effects. Therefore, in order to clarify this we have included a sentence in “Data analysis” section indicating that “all covariates were taken as fixed effects.”

Line 180-182: What did you do about collinearity? You need to explain more here.

- Thank you, Only uncorrelated covariates (p value >0.05) were considered together in the models. Now this is also explained in the Data analysis section.

Line 183-184: More information is needed here. How was model selection conducted. How did you conduct model selection? Did you compare all possible model structures? Did you try interactions? Did you use the dredge function in MuMIn? Please explain your process.

- Thank you again. Now we specify that best models were those with the lowest AIC. Regarding to model structures, now we specify this: “Only model structures making ecological sense according to the species-specific diet and foraging strategies were compared”. In the first paragraph of Data Analysis section, in the sentence “GLMs were used to analyze factors affecting the egg volume” we have also added “as well as their statistical interactions”. Dredge function in MuMin was not used for model selection.

Results

It will be important to see the data behind the relationships suggelsted in your top models. I want to see the relationship between egg-volume and and your predictors. For example, what does the egg-volume look like for eggs measured when wave height is high and discards are high vs low wave height and low discards vs low wave height and high discards? You need to report your actual model results somewhere. The top model for each species will be fine, but it is needed to evaluate your conclusions!

- Thanks, we have now added three more tables in the supporting information. These tables show annual values of the observed egg volume and covariates retained by the best explanatory model for each species (See Tables S9, S10 and S11).

Line 216-218: You need to explain what “pretending variables” are.

- Thanks again, we have added a sentence explaining what pretending variables are.

Discussion

Discussion would be clearer if separated by species or by large versus small scale effects. Headings would also be helpful if allowed in journal format.

- Thank you for this observation. In fact, this was our first option for which in addition many hours were dedicated to this purpose. However, when finished, most of authors agreed that the reading was much more difficult to follow and we decided to change it to the current format. We all came to the conclusion that this format works well for other sections of the manuscript such as Methods or Results, but not for the Discussion section. Although we do not recommend it, we can change it if necessary.

Line 232 -236: If for all three species egg- volume was better predicted by including both local and large-scale variables, how can you say the relative importance was different? How did you evaluate this? Is the magnitude of the effect sizes different? If you mean that the effects of the same variable were different among species than you need to say this instead.

- Thanks for the comment. We have changed this sentence accordingly as follows:

“The effects of the wNAO and the local variables measured in spring were different for each study species.”

Line 239-241: Please explain the importance of this effect with respect to the slow life-history of the shearwaters. Why would we expect a carry-over effect in a species with a slow-life history? Especially with respect to reproductive investment. Are shearwaters income or capital breeders?

- We have added the following sentence: “This result also suggests that Cory's Shearwaters act as capital breeders, however further research is needed in this regard, so this possibility should be taken with caution.”

Line 258-259: Please explain the logic linking this sentence to the topic of this paragraph?

- Thank you, we have modified this sentence to put it on the context of the paragraph.

Line 263: Is it possible your variables are a poor proxy for food abundance?

- We agree they could be better (e.g. using real amounts of discards), but this type of data was not available for the studied period in the Western Mediterranean. However, the choice of these proxies was not made at random. Considering the high proportion of fish discarded related to the total catch, (especially in the studied period), it makes sense to think that the more trawlers are in the area, the more discards are generated. Regarding natural prey, what we did first was to find out the diet of each species according to scientific literature. Then, from among all prey items we considered those that are most abundant in the study area. Finally, we calculated annual estimates of the abundance of these species. Another potentially important factor is that the study area is a very rich area in terms of prey abundance, suggesting that the abundance of food is not a limiting factor and therefore, detectability and accessibility could be more important.

Line 270: Can you provide better support for this possibility? What are the main prey sources of the shearwaters and is there evidence they are affect by larger waves?

- According to our methodology (see above), for shearwaters, sardines are among the most abundant natural prey in the study area. We have made a second search but we have not found more references supporting the possibility that larger waves may drive small pelagic to deeper waters (apart from Dunn 1973).

Line 277: Reference software error on this line?

- Yes, this error only appears in the draft pdf manuscript. We have changed the reference format so it should be fixed.

Line 291-293: Other possible explanations here?

- There is a very clear density dependence effect on the egg volume of Audouin's gulls because the study area hosts up to 73% of the total world population. Regarding the other two species, perhaps the density dependence has almost no or very little effect on the egg volume.

Conclusions

My main feedback for the conclusions is to change the language so it is less certain. For example replace uses of the word influence with correlated. I think it is important to remember all the results are correlations and do not necessarily concretely prove the conclusions described here.

- Thanks, we have changed the term ‘influence’ by ‘are correlated’

Figure caption three is confusing. Are these predictions from the best models for each species or are these the predictions from models with both spring and winter conditions? Please clarify.

- This figure show; i) the egg volume observed, ii) the egg volume predicted by the best explanatory model (which includes winter and spring conditions), and the wNAO alone. We have eliminated “that consider both, winter- and spring-conditions “ from the figure legend for better understanding.

---

## [Decision Letter · Decision Letter 1]

16 Mar 2022

PONE-D-21-28485R1It’s not all abundance: detectability and accessibility to food also explain breeding investment in long-lived animalsPLOS ONE

Dear Dr. Real,

Thank you for submitting your manuscript to PLOS ONE. After careful consideration, we feel that it has merit but does not fully meet PLOS ONE’s publication criteria as it currently stands. Therefore, we invite you to submit a revised version of the manuscript that addresses the points raised during the review process.

We look forward to receiving your revised manuscript.

Kind regards,

Vitor Hugo Rodrigues Paiva, Ph.D.

Academic Editor

PLOS ONE

Reviewers' comments:

Reviewer's Responses to Questions

**Comments to the Author**

1. If the authors have adequately addressed your comments raised in a previous round of review and you feel that this manuscript is now acceptable for publication, you may indicate that here to bypass the “Comments to the Author” section, enter your conflict of interest statement in the “Confidential to Editor” section, and submit your "Accept" recommendation.

Reviewer #3: All comments have been addressed

Reviewer #4: (No Response)

2. Is the manuscript technically sound, and do the data support the conclusions?

Reviewer #3: Yes

Reviewer #4: No

3. Has the statistical analysis been performed appropriately and rigorously? 

Reviewer #3: Yes

Reviewer #4: No

4. Have the authors made all data underlying the findings in their manuscript fully available?

Reviewer #3: Yes

Reviewer #4: Yes

5. Is the manuscript presented in an intelligible fashion and written in standard English?

Reviewer #3: Yes

Reviewer #4: Yes

6. Review Comments to the Author

Reviewer #3: Thank you for your careful consideration and response to all comments. I appreciate it! I think this paper has been greatly improved and presents some interesting ideas that can be tested for more species!

Reviewer #4: This is an interesting study that examines the correlation between egg volume of three different seabird species in relation to proxies of food abundance, accessibility and detectability. However, it is difficult to judge from this manuscript whether these chosen proxies well represent the processes the authors intended to test (see details below). In addition, the lack of hypotheses and predictions of changes in eggs volume as a function of differences in life history strategies, feeding strategies, and population sizes per species, make the results difficult to follow, and the discussion is for me too focused on the studied species, without really drawing conclusions in a broader context.

In addition, I am concerned about the way the analyses were conducted and at least presented. This is surely also due to the fact that the available data do not contain any description (e.g. with a readme.txt file) and that no R codes were associated with them. So I attached with this review an R code in which I tried to reproduce the best model for the shearwater dataset. It seems to me that instead of using GLMs, the authors actually simply performed a multiple linear regression (see detailed comments), and that the R-squared of the best model seems to be really low (<3%), which question the robustness of the results (at least for the shearwater data, and I would urge the authors to present R squared for the others species as well). Moreover, the authors did not present the fit of the models (relationships between egg volume and covariates for the best models) and did not say whether the basic assumptions were verified (linearity, normality, homogeneity independence), which also makes it difficult to judge the quality of the fit of their results. Instead the authors seem to have used an inappropriate formula, originally adapted to survival models made for capture history data (L190-191). This type of data does not seem to be part of the study design. In case I am wrong, this would mean the entire Methods section should be completely rewritten to better explained how modeling where conduced. I suggested R packages and alternatives approaches to analyze their data, to assess goodness of fit and the relative importance of each predictor (R code attached and details comments below). I also questioned whether mixed effects models could/should be used to control for potential inter-annual variability.

In conclusion, although I must say that the conclusion brought by the authors, of the importance of availability and accessibility of resources is very interesting to me, I was unfortunately not convinced by the analyses, and I hope the R code that I shared could be helpful to improve the manuscript. Below I provide detailed comments that I hope will be helpful:

Title

L1: “accessibility of” instead of “to” I think

L3: Given the introduction and especially the discussion very focused on seabirds I would change “animals” to “seabirds” instead, or at least to “marine animals” or “marine predators”

Abstract

L20: “long-term” please specify how many years (at least in average) directly in the abstract, please also specify studied regions, species names and sample size.

L24: “competition” please specify inter- and or intra-specific competition?

L26: “predictive power” estimated with cross-validation was not preformed in this study, perhaps “goodness of fit” instead?

L30: please specify if “detectability” corresponds to “foraging conditions” and “accessibility” to “competition” (by splitting the bracket). If it is not the case, please consider to add in bracket that both are influenced by same processes (foraging conditions + competition)

Introduction

L54: “marine” top predators would be better I think

L67: more details of different “life histories” traits would required in my opinion

L80-83: “we assess the influence of i) [...] ii) [...] as predictors of...” please reformulate, this sentence is not correct

L83-84: please provide here a clear definitions of detectability and accessibility directly in the introduction, as they are key concepts of the study. In my opinion, they are not interchangeable, and therefore should also be considered separately in the text, whereas they are always used a combination (e.g. L136, L211, L268, L272, L274, L281). I invite the authors to cite Matthiopoulos 2003, a seminal article on accessibility in ecology.

L85-87: There is a clear lack of predictions/expectations of how effects on egg volume should change differently depending on species foraging strategies, life histories, and population densities. One idea would be to summarize these in a summary table...

Methods

L98: “temporal variance” it is not clear to me how variance will be then used in the analyses. From the data and results, only the average egg volume seems to have been analyzed, but not the change in variance over time?

L104: “eggs volume were measured” this was done before calculating “temporal variance” I guess (cf. Precedent comment)? If yes, please consider re-ordering

L109: I would have deleted it if it wasn't “necessary”, plus the paragraph on ethics right after is already long enough.

L112-119: This ethic paragraph could shorten, I think.

L120: “Predictors” the term “covariates” is also used later, it would be better for the reader to choose a term and remain consistent in the manuscript.

L130: Paragraph title could be renamed as “Accessibility vs. Detectability” for clarity (with a subtitle perhaps)

L131: I don't quite see what these "physical barriers" could be that could for example prevent access but not detection? It can also be argued that birds must first access foraging areas based on, for example, winds, distances to colonies, and forager densities, and then, on a finer scale, engage in a area restricted search (ARS), where detectability plays a more important role, based on, for example, water turbidity, wave height etc (see Weimerskirch 2007).

L136: “detectability and accessibility” it seems to be considered interchangeable here, whereas they are two distinct processes with different scales and potentially influenced by different factors. More general and theoretical references could be cited here.

L139: It is very nice to investigate cumulative effects! We can also look at the variance of the covariates instead of the mean values (cf. Stenseth et al 2002)

L146: Why “early” is not clear to me, this should be defined in the ‘Field data’ paragraph

L148: “breeding pairs numbers” could be a poor indicators of inter- and intra-specific competition, as densities at sea can vary according to many factors (social behavior, environmental productivity, variability and predictability, etc) it might be better idea to use distance to the colony as a proxy of competition (cf. Wakefield et al. 2011,

L159: Does “Per capita” refer to per individual, per species, per fisheries boat? Please reformulate

L163: The "CPUE" is actually a proxy of prey available to seabirds, but this would need better justification (i.e., reference materials) because it seems to me that what is caught by fisheries does not necessarily reflect what is available to seabirds, unless one assumes that catches are proportional to fish stocks and that fisheries and seabirds forage in the same areas (in which case the assumptions would need to be clearly stated)

L170: Why “horsepower” could be considered as a good proxy for fisheries discarded should be clearly explained (readers should understand without looking a the reference is not enough).

L170: “Egg volume” and not its “temporal variance” is analyzed instead of what it is said L98.

L178: “generalized linear models” the family used to model egg volume is not given. I am assuming that the gaussian family with an identity link function. In that case the authors in fact used a simple (multiple) linear regression and not a GLM (or maybe a GLS if they used more complex variance structure cf. Zuur et al. 2009)

L179: “modal clutch” please define what “modal” means

L179-181: This sentence is completely unclear to me. Did you standardized all variables using center=T and scale=T with the scale function in R (so that means=0 and standard deviation=1)? Dividing by 1000 is not necessary in my opinion. Also, I tried to fit the best model for the shearwater using the data available online, and to be able to retrieve the same estimates presented in supplementary (Table S12), I had to use the unscaled data (the raw data, please see attached R code). Could the authors clarified this point?

L183: “collinearity” Variance Inflation Factors should also be investigated (cf. Zuur et al. 2009)

L185: The authors should explained more (by making clear predictions in the introduction) what make “ecological sense” as models tables as they stands look the results of arbitrary choices.

L189: “The proportion of total annual variance” is unclear to me, the formula would required a reference from a statistical paper or book.

L190-191: The terms "constant model" and "time-dependent model" make no sense to me here, it seems that the authors are trying to use a survival model with a notation made for capture history data. If I have misunderstood, much more detail should be provided on the design of the study (are the marked and known birds or nests monitored each year? see how the data are presented, for example, in Harris et al. 2005).

L191-192: Since I assume the authors used a GLM with a Gaussian distribution, this is equivalent to a linear model for me, so this formula doesn't make sense to me here. And the R-squared should be provided. The relative proportion of variance explained by each predictor for the best model could then be given, in order to assess their relative importance (cf. rwa or relaimpo R packages, see Grömping 2006).

L192: The classical diagnostic plots to check for models assumptions should be provided in supplementary material and or at least to be said to have been checked. Here the list of assumptions (also worked for GLMs):

1.Linearity of the data

2.Normality of residuals

3.Homogeneity of residuals variance

4.Independence of residuals error terms

L193: It is particularly important to test the linearity of the relationships, because nonlinear relationships are widely reported in ecology (such as the bell shape) and easily verified (by the use of polynomials or generalized additive models).

L193: As I understand it, bird identity is not available for the study, which would explain why mixed-effects models cannot be used. However, given that several data are nested by year, I wonder if at least the year should not be used as a random structure (it seems that several studies on egg size change have done this, e.g. Kvalnes et al. 2013, Verhoeven et al. 2019).

Results:

L198: Figure 3 presents the time series of egg volume whereas the we are expecting to see how the drivers are correlated to egg volume. Outputs of best models should thus be given (I would encourage the authors to look at the R car package and the avPlots function, and the ggeffects R package with ggpredict function, please see R code and references attached).

L199: “decreased the egg volume” this is a too strong assertion since this is only based on correlation, it is very likely that these are the result of indirect effects.

L202: “separately” this corresponds to which model ? (model 4 for wNOA I guess, but I cannot find local condition only models?) I would rather use the best model and to partition the variance per predictor.

L202: “total annual variance” I did not understood how authors managed to calculate this, in case of GLM the proportion of explained variance should be 1-deviance/null deviance. But here as gaussian regressions have been used, the R-squared should be reported. I did fit the best shearwater model with the data available and the R-squared is rather low with only 3% (see code attached), I also tried to plot the regression lines, and it appears that slopes of the relationships looks very weak. I am thus wondering if the effect found by the authors is rather anecdotal, and I would encourage to explore new covariates.

Table 1: I would be more interesting to me to present effect sizes of the models (estimates e.g. Table S12) rather than (incomplete) model selection of Tables 1, 2, and 3. I think providing complete model selection tables in supplementary is already enough. Pvalues or at least 95% Confidence Intervals should be provided however for the estimates tables, with R-squared (e.g. Table S12).

Table S2: few models were below dAIC<2 and thus model averaging should be conduced I think for these Sandwich tern models.

Discussion

In my opinion the Discussion is too focused on the results and studied species, and therefore failed to generalize over a broader range of species, systems and life histories. There is also a lack of comparisons with studies that also investigated changed in egg volume such as Barrett et al. 2012, Bennett et al. 2017, Tomita et al. 2009, Verhoeven et al. 2019. These results are only correlative, could the authors think and suggest ways / experiments to answer this drawback?

Data

No readme.txt, and no codes, which make the analyses hardly reproducible. For example are the variables already scaled? What are the units of each variable? What family and link function were used for the GLMs? How models within models selection table were chosen?

References mentioned in this review

Barrett, R., Nilsen, E. & Anker-Nilssen, T. (2012). Long-term decline in egg size of Atlantic puffins Fratercula arctica is related to changes in forage fish stocks and climate conditions. Mar. Ecol. Prog. Ser., 457, 1–10.

Bennett, J.L., Jamieson, E.G., Ronconi, R.A. & Wong, S.N.P. (2017). Variability in egg size and population declines of Herring Gulls in relation to fisheries and climate conditions. ACE, 12, art16.

Catry, P., Lemos, R.T., Brickle, P., Phillips, R.A., Matias, R. & Granadeiro, J.P. (2013). Predicting the distribution of a threatened albatross: The importance of competition, fisheries and annual variability. Progress in Oceanography, 110, 1–10.

Grömping, U. (2006). Relative Importance for Linear Regression in R : The Package relaimpo. J. Stat. Soft., 17.

Harris, M., Anker-Nilssen, T., McCleery, R., Erikstad, K., Shaw, D. & Grosbois, V. (2005). Effect of wintering area and climate on the survival of adult Atlantic puffins Fratercula arctica in the eastern Atlantic. Mar. Ecol. Prog. Ser., 297, 283–296.

Kvalnes, T., Ringsby, T.H., Jensen, H. & Sæther, B.-E. (2013). Correlates of egg size variation in a population of house sparrow Passer domesticus. Oecologia, 171, 391–402.

Matthiopoulos, J. (2003). The use of space by animals as a function of accessibility and preference. Ecological Modelling, 159, 239–268.

Tomita, N., Niizuma, Y., Takagi, M., Ito, M. & Watanuki, Y. (2009). Effect of interannual variations in sea-surface temperature on egg-laying parameters of black-tailed gulls (Larus crassirostris) at Teuri Island, Japan. Ecol Res, 24, 157–162.

Verhoeven, M.A., Loonstra, A.H.J., McBride, A.D., Tinbergen, J.M., Kentie, R., Hooijmeijer, J.C.E.W., et al. (2020). Variation in Egg Size of Black-Tailed Godwits. Ardea, 107, 291.

Wakefield, E.D., Phillips, R.A., Trathan, P.N., Arata, J., Gales, R., Huin, N., et al. (2011). Habitat preference, accessibility, and competition limit the global distribution of breeding Black-browed Albatrosses. Ecological Monographs, 81, 141–167.

Weimerskirch, H. (2007). Are seabirds foraging for unpredictable resources? Deep Sea Research Part II: Topical Studies in Oceanography, 54, 211–223.

7. PLOS authors have the option to publish the peer review history of their article (what does this mean?). If published, this will include your full peer review and any attached files.

Reviewer #3: No

Reviewer #4: No

---

## [Author Response · Author response to Decision Letter 1]

28 Jul 2022

Reviewer #3: Thank you for your careful consideration and response to all comments. I appreciate it! I think this paper has been greatly improved and presents some interesting ideas that can be tested for more species!

The authors: Thank you for these words and for your constructive comments.

Reviewer #4: This is an interesting study that examines the correlation between egg volume of three different seabird species in relation to proxies of food abundance, accessibility and detectability. However, it is difficult to judge from this manuscript whether these chosen proxies well represent the processes The authors intended to test (see details below).

The authors: We thank the reviewer for the comment but that was not the original purpose of our study. The main goal was to assess whether local variables are better predictors than a large-scale climatic index (the North Atlantic Oscillation) of the interannual variations of egg volume (see the Abstract). It is important to highlight that we have not correlated these each covariate with the egg volume. We have built models with a combination of them to predict the interannual variation of the mean egg volume depending on environmental conditions considering different spatial scales.

In addition, the lack of hypotheses and predictions of changes in eggs volume as a function of differences in life history strategies, feeding strategies, and population sizes per species, make the results difficult to follow, and the discussion is for me too focused on the studied species, without really drawing conclusions in a broader context

The authors: Many studies assume that the correlation between large-scale indexes and ecological processes is due to the effect of local variables. However, the relative role of these two type of covariates (large-scale indexes vs local conditions) has seldom been investigated. Our starting hypothesis was that local variables are better predictors of the mean egg volume variability than large-scale climatic indices as often assumed. We wrote that “Given the smaller area used by seabirds during the breeding period relative to the winter distribution, we expect local variables to be a better predictor than the North Atlantic Oscillation index on egg volume” to clarify this point (see L85-87).

In addition, I am concerned about the way the analyses were conducted and at least presented. This is surely also due to the fact that the available data do not contain any description (e.g. with a readme.txt file) and that no R codes were associated with them.

The authors: We agree. All statistical procedures are described in the Methods section but we are happy to share our R-scripts if necessary. We have now added three scripts (one per species) to run all GLM.

So I attached with this review an R code in which I tried to reproduce the best model for the shearwater dataset.

The authors: Thank you for this. We appreciate it. We have added the scripts and they are now available for review.

It seems to me that instead of using GLMs, The authors actually simply performed a multiple linear regression (see detailed comments),

The authors: This is in part correct multiple regressions are a particular case of linear models in wich error structure is considered normally distributed. The main difference is that parameters in multiple regressions are estimated using least of squares methods while here we used likelihood based procedures. We now provided the R-scripts to run the whole analysis.

and that the R-squared of the best model seems to be really low (<3%), which question the robustness of the results (at least for the shearwater data, and I would urge The authors to present R squared for the others species as well).

The authors: This was a misunderstanding. In general, egg volume variability in birds depends on genetic effects (female size; >80%), approximately a 9% depends on food intake and around 11% depends on other factors (see Christians 2002 cited in the text). What we assessed is how the mean egg volume changes over the years and how much of this variation is due to environmental factors (large-scale vs local). For example, if the total volume of an egg is 75 cm3, approximately 60 Cm3 depends on the size of the female, the remaining 15cm3 depends on environmental conditions (e.g. food intake). We aimed to explain the inter-annual variations of those 15cm3 considering environmental conditions measured at different spatio-temporal scales. Results show that there is a strong correlation between the egg volume observed and the egg volume predicted by the models (see e.g. Figures 3, S15, S16 and S17). R-squared are all above the 0.59.

Moreover, The authors did not present the fit of the models (relationships between egg volume and covariates for the best models)

The authors: We are not sure to have understood this comment. We assessed the fit between the mean egg volume predicted and the mean egg volume observed, which is 59-79% depending on the species (see Results section and S15, S16 and S17).

and did not say whether the basic assumptions were verified (linearity, normality, homogeneity independence), which also makes it difficult to judge the quality of the fit of their results.

The authors: We agree. We have now added more information on model performances in the manuscript. The tests used to check these assumptions can be also found in the R-scripts.

Instead The authors seem to have used an inappropriate formula, originally adapted to survival models made for capture history data (L190-191). This type of data does not seem to be part of the study design. In case I am wrong, this would mean the entire Methods section should be completely rewritten to better explained how modeling where conduced. I suggested R packages and alternatives approaches to analyze their data, to assess goodness of fit and the relative importance of each predictor (R code attached and details comments below). I also questioned whether mixed effects models could/should be used to control for potential inter-annual variability.

The authors: We have used these formulas to assess the proportion of the temporal variance explained by the best models. On the other hand, the correlations shown in figures S15, S16 and S17 also support these results.

In conclusion, although I must say that the conclusion brought by The authors, of the importance of availability and accessibility of resources is very interesting to me, I was unfortunately not convinced by the analyses, and I hope the R code that I shared could be helpful to improve the manuscript. Below I provide detailed comments that I hope will be helpful:

Title

L1: “accessibility of” instead of “to” I think

The authors: We have changed that.

L3: Given the introduction and especially the discussion very focused on seabirds I would change “animals” to “seabirds” instead, or at least to “marine animals” or “marine predators”

The authors: Thank you, we agree and we have changed it accordingly.

Abstract

L20: “long-term” please specify how many years (at least in average) directly in the abstract, please also specify studied regions, species names and sample size.

The authors: We agree. However the Abstract already has 225 words and we would like to stay well within the limits fixed by the journal. These details can be found in the Methods section.

L24: “competition” please specify inter- and or intra-specific competition?

The authors: This information is also detailed in the Methods section. We believe that including this information in the Abstract would not improve it.

L26: “predictive power” estimated with cross-validation was not preformed in this study, perhaps “goodness of fit” instead?

The authors: We assume the reviewer refers to the use of cross-validation to predict the error in Generalized Linear Models. Here we refer to the predictive power of the models.

L30: please specify if “detectability” corresponds to “foraging conditions” and “accessibility” to “competition” (by splitting the bracket). If it is not the case, please consider to add in bracket that both are influenced by same processes (foraging conditions + competition)

The authors: In this sentence we considered that foraging conditions may influence both, detectability and accessibility of food. However, we agree that competition has more to do with food accessibility so we have changed this sentence.

Introduction

L54: “marine” top predators would be better I think

The authors: We agree and we have added “marine” to the sentence. However, in this study, we use large-scale climatic indices and long-term data on seabirds' egg volume. For this reason, it is also very important to keep in mind that these seabirds are “long-lived and long-ranged marine top predators”.

L67: more details of different “life histories” traits would required in my opinion

The authors: In this study, we do not use data on ecological parameters at the individual level, because they were not fully available. However, some species-specific life-history traits that are important for the study are provided through the manuscript and the supporting information (see e.g. Table S4).

L80-83: “we assess the influence of i) [...] ii) [...] as predictors of...” please reformulate, this sentence is not correct

The authors: We agree and we have modified the sentence accordingly.

L83-84: please provide here a clear definitions of detectability and accessibility directly in the introduction, as they are key concepts of the study.

The authors: Detectability and accessibility are indeed key concepts. We described these terms in the Methods section when we refer them to the covariates considered.

In my opinion, they are not interchangeable, and therefore should also be considered separately in the text, whereas they are always used a combination (e.g. L136, L211, L268, L272, L274, L281).

The authors: We agree. We used these terms together throughout the text because some of the covariates used in the models influence both, detectability and accessibility of food. We do not mean that they are interchangeable. Simply they should be ‘considered’ together.

I invite The authors to cite Matthiopoulos 2003, a seminal article on accessibility in ecology.

The authors: Thank you for the reference. We have cited this article in the manuscript and added it to the list.

L85-87: There is a clear lack of predictions/expectations of how effects on egg volume should change differently depending on species foraging strategies, life histories, and population densities. One idea would be to summarize these in a summary table...

The authors: We agree. These issues have been discussed in de Discussion section. See also Tables S4 and S5.

Methods

L98: “temporal variance” it is not clear to me how variance will be then used in the analyses. From the data and results, only the average egg volume seems to have been analyzed, but not the change in variance over time?

The authors: The change in egg volume in relation to an average value has a clear temporal components. Here by temporal variance we refer to the changes of the mean egg volume over the years. In general, less than 9% of the egg volume in birds can be explained by food (See e.g. Christians 2002) as most of the total egg volume (around 80%) depends on genetic-related factors (e.g. female size) which here we were not be able to account for. Our aim is not to assess the influence that covariates have on the total egg volume. We use models to assess how the proportion of the egg volume that depends on environmental factors (the 9% mentioned above) changes over the years.

L104: “eggs volume were measured” this was done before calculating “temporal variance” I guess (cf. Precedent comment)? If yes, please consider re-ordering

The authors: See comments above.

L109: I would have deleted it if it wasn't “necessary”, plus the paragraph on ethics right after is already long enough.

L112-119: This ethic paragraph could shorten, I think.

The authors: We agree. This is because of the many government institutions that have to be mentioned.

L120: “Predictors” the term “covariates” is also used later, it would be better for the reader to choose a term and remain consistent in the manuscript.

The authors: We partially agree. The term covariates refers to the set of values of one or more (co)variables that are used as predictors (idependent variables) of the egg volume. In this case “predictors” can refer to a set of covariates and it is used to inform the reader that we used a combination of several covariates. Technically speaking a predictor can be an interaction effect, while a covariates are only a single variable (rather than a combination of two or more covariates). This difference depends of course on the sentence. For example the term predictors in the title “Predictors of the egg volume” is more informative than “Covariates of the egg volume”.

L130: Paragraph title could be renamed as “Accessibility vs. Detectability” for clarity (with a subtitle perhaps)

The authors: This would not be fully correct because accessibility also involves competition, which is described in another section.

L131: I don't quite see what these "physical barriers" could be that could for example prevent access but not detection?

The authors: There are many examples. The wave height for example can reduce the number of successful attempts in Sandwich terns (see also references in the text).

It can also be argued that birds must first access foraging areas based on, for example, winds, distances to colonies, and forager densities, and then, on a finer scale, engage in a area restricted search (ARS), where detectability plays a more important role, based on, for example, water turbidity, wave height etc (see Weimerskirch 2007).

The authors: When we use the term "accessibility", we are not only referring to the foraging areas but also to the food itself. According to our approach, seabirds must first detect food (prey) and then try to access it.

L136: “detectability and accessibility” it seems to be considered interchangeable here, whereas they are two distinct processes with different scales and potentially influenced by different factors. More general and theoretical references could be cited here.

The authors: See the answer to the similar comment above. These two terms are different of course, but sometimes a given covariate can influence both.

L139: It is very nice to investigate cumulative effects! We can also look at the variance of the covariates instead of the mean values (cf. Stenseth et al 2002)

The authors: We agree and it is an interesting comment. We would like to investigate the variance as well as the predicted trajectories of the covariates provided by current scenarios of future oceanographic changes. However, we will leave this interesting aspect for a future work.

L146: Why “early” is not clear to me, this should be defined in the ‘Field data’ paragraph

The authors: We said ‘early’ because this is the period when seabirds mobilize reserves used in the formation of eggs but we ignore the exact timing for each species or mechanisms involved. We used a general term to mean ‘before’.

L148: “breeding pairs numbers” could be a poor indicators of inter- and intra-specific competition, as densities at sea can vary according to many factors (social behavior, environmental productivity, variability and predictability, etc) it might be better idea to use distance to the colony as a proxy of competition (cf. Wakefield et al. 2011,

The authors: We agree. Competition and behaviourl aspects of foraging at sea are interesting issues to explore. Unfortunately we do not have enough telemetry data for each species to investigate this point.

L159: Does “Per capita” refer to per individual, per species, per fisheries boat? Please reformulate

The authors: The Latin meaning of the term “per capita” is “per head”. Therefore, it refers to individuals.

L163: The "CPUE" is actually a proxy of prey available to seabirds, but this would need better justification (i.e., reference materials) because it seems to me that what is caught by fisheries does not necessarily reflect what is available to seabirds, unless one assumes that catches are proportional to fish stocks

The authors: Catch per unit of effort (CPUE) is an indirect measure of the abundance of a target species widely used in fishery stock assessment. Using a proxy like this is the best option in studies that consider large temporal and spatial scales.

and that fisheries and seabirds forage in the same areas (in which case the assumptions would need to be clearly stated)

The authors: Datasets used to estimate CPUE correspond to the same areas were seabirds forage during the breeding season when foraging areas are more restricted (see Methods section).

L170: Why “horsepower” could be considered as a good proxy for fisheries discarded should be clearly explained (readers should understand without looking a the reference is not enough).

The authors: We agree. Now we have added more details in this regard.

L170: “Egg volume” and not its “temporal variance” is analyzed instead of what it is said L98.

The authors: We agree. Thank you for pointing this out. There is a mistake in L98. We have now changed the sentence “we recorded the temporal variance of the annual mean volume of the modal clutch” by “we recorded the annual mean volume of the modal clutch”

L178: “generalized linear models” the family used to model egg volume is not given. I am assuming that the gaussian family with an identity link function. In that case the authors in fact used a simple (multiple) linear regression and not a GLM (or maybe a GLS if they used more complex variance structure cf. Zuur et al. 2009).

The authors: Correct. We used Gaussian family and identity link. The multiple linear regressions (MLR) are indeed a particular case of GLM. However in MLR parameters are least-squares estimates. See the answer to a similar comment above. In this case the two terms are correct, but GLM also inform on how parameters are estimated.

L179: “modal clutch” please define what “modal” means

The authors: We agree and we have now improved this sentence accordingly.

L179-181: This sentence is completely unclear to me. Did you standardized all variables using center=T and scale=T with the scale function in R (so that means=0 and standard deviation=1)?

The authors: No, we first scaled covariates (only those needing to be scaled) by dividing by 1000. Then we centered these variables by substracting the mean to each value which is equivalent to the using “center=T” in R. Now we have modified this sentence for better understanding.

Dividing by 1000 is not necessary in my opinion.

The authors: Using the function “scale=T” is only another way to scale data.There are many ways to scale data. Dividing by 1000 is simply another way to do it.

Also, I tried to fit the best model for the shearwater using the data available online, and to be able to retrieve the same estimates presented in supplementary (Table S12), I had to use the unscaled data (the raw data, please see attached R code). Could the authors clarified this point?

The authors: We think the reason you have to downscale the data to get the same results is that the data available online has already been scaled. Now the original scripts are also available for review. Observe that when the name of our covariates start with the term "ScalStd..." like e.g. "ScalStdTrawlHp" or "ScalStdCPUE_Sardine", this means that these covariates have already been scaled and standardized.

L183: “collinearity” Variance Inflation Factors should also be investigated (cf. Zuur et al. 2009)

The authors: We agree. We used cross correlations to assess for collinearity and we have not considered together correlated covariates in the same model. Now we have also performed a VIF analysis for best models as suggested. Considering a threshold value of 3 as in Zuur et al. 2010, and there is no evidence of collinearity as in the cross-correlation methods. VIF analysis are also included in the scripts. In models with interactions, interaction terms were also centered (using center=T) as suggested by Robinson and Schumacker 2009. We have also added an explanation of this procedures in the manuscript as follows:

“For the best models, we also run tests to check the variance inflation factors (VIFs). Only models were all VIF values were <3 were considered [48]. To check VIFs in models with interaction terms we first centered these covariates as suggested by [49].”

We have also added these two references:

48. Zuur, A. Leno, E.N. and Elphick, C.S. A protocol for data exploration to avoid common statistical problems. Methods Ecol. Evol. 2010; 1: 3-14. http://www.respond2articles.com/MEE/

49. Robinson, Cecil & Schumacker, Randall. (2009). Interaction Effects: Centering, Variance Inflation Factor, and Interpretation Issues. Multiple Linear Regression Viewpoints. 35.

L185: The authors should explained more (by making clear predictions in the introduction) what make “ecological sense” as models tables as they stands look the results of arbitrary choices.

The authors: We disagree. Models have been selected using realistic hypotheses on what may influence ecological process or species-specific characteristics. We wrote “…ecological sense according to species-specific diet and foraging strategies”.

L189: “The proportion of total annual variance” is unclear to me, the formula would required a reference from a statistical paper or book.

The authors: Please, see comments above and the last paragraph of the Methods section.

L190-191: The terms "constant model" and "time-dependent model" make no sense to me here, it seems that the authors are trying to use a survival model with a notation made for capture history data. If I have misunderstood, much more detail should be provided on the design of the study (are the marked and known birds or nests monitored each year? see how the data are presented, for example, in Harris et al. 2005).

The authors: We partially agree. Model notation (and model description) is in many cases subjective. The important point is that notation is explained for the readers to understand the models. We did it here and we used a notation similar to the GLM syntax for those reader familiar with it. Also note that in the suggested reference by Harris et al. 2005 models are described as “..constant and fully time-dependent models” (pag 291). This description refers indeed to survival models which are logistic regressions (another particular case of GLM).

L191-192: Since I assume the authors used a GLM with a Gaussian distribution, this is equivalent to a linear model for me, so this formula doesn't make sense to me here. And the R-squared should be provided. The relative proportion of variance explained by each predictor for the best model could then be given, in order to assess their relative importance (cf. rwa or relaimpo R packages, see Grömping 2006).

The authors: See the answer to a similar comment above about GLM vs linear regression. We assessed the fit using predicted vs observed values. Equivalences of R-squares are given in the text.

L192: The classical diagnostic plots to check for models assumptions should be provided in supplementary material and or at least to be said to have been checked. Here the list of assumptions (also worked for GLMs):

1.Linearity of the data

2.Normality of residuals

3.Homogeneity of residuals variance

4.Independence of residuals error terms

The authors: We agree. We have now modified the text. The functions used for model diagnostics are also found in the scripts.

L193: It is particularly important to test the linearity of the relationships, because nonlinear relationships are widely reported in ecology (such as the bell shape) and easily verified (by the use of polynomials or generalized additive models).

The authors: We agree. Idem to previous response.

L193: As I understand it, bird identity is not available for the study, which would explain why mixed-effects models cannot be used. However, given that several data are nested by year, I wonder if at least the year should not be used as a random structure (it seems that several studies on egg size change have done this, e.g. Kvalnes et al. 2013, Verhoeven et al. 2019).

The Authors: We agree in theory, however “year” as a fixed factor was significant and this was the variability we wanted to explain using the covariates. Having both types of variables (year and covaraites) into the model was redundant.

Results:

L198: Figure 3 presents the time series of egg volume whereas the we are expecting to see how the drivers are correlated to egg volume. Outputs of best models should thus be given (I would encourage the authors to look at the R car package and the avPlots function, and the ggeffects R package with ggpredict function, please see R code and references attached).

The Authors: Figure 3 shows the observed values of egg-volume over time and those predicted by the best model. In this sense it can be seen as a graphical illustration of the fit of the model. This is explained in the figure text.

L199: “decreased the egg volume” this is a too strong assertion since this is only based on correlation,

The Authors: The model constrains the expected value to a linear trend. Whether this result an increasing or decreasing trend depends on the sign of the linear predictor. In this case it was predicted a decreasing trend based on statistical inference.

it is very likely that these are the result of indirect effects.

The Authors: We disagree. Several studies have shown that wave height has a direct and statistically significant effect on seabirds foraging success (see reference list).

L202: “separately” this corresponds to which model ? (model 4 for wNOA I guess, but I cannot find local condition only models?)

The Authors: We agree. Now we have included these three models (one per species) with local conditions only (see Model 3 in Tables 1 and S1; Model 10 in Tables 2 and S2 and Model 4 in Tables 3 and S3).

I would rather use the best model and to partition the variance per predictor.

The Authors: We understand. It is a slightly different way to show the results. We did specify the variance explained by the different set of predictos (local vs large-scale). We think it is a more direct way to inform readers, considering the starting hypotheses.

L202: “total annual variance” I did not understood how authors managed to calculate this, in case of GLM the proportion of explained variance should be 1-deviance/null deviance. But here as gaussian regressions have been used, the R-squared should be reported. I did fit the best shearwater model with the data available and the R-squared is rather low with only 3% (see code attached), I also tried to plot the regression lines, and it appears that slopes of the relationships looks very weak. I am thus wondering if the effect found by the authors is rather anecdotal, and I would encourage to explore new covariates.

The Authors: See answer to a similar comment above. We had explained the rational and each step of the analysis in the manuscript and in the supplementary information for a better understanding.

Table 1: I would be more interesting to me to present effect sizes of the models (estimates e.g. Table S12) rather than (incomplete) model selection of Tables 1, 2, and 3. I think providing complete model selection tables in supplementary is already enough. Pvalues or at least 95% Confidence Intervals should be provided however for the estimates tables, with R-squared (e.g. Table S12).

The Authors: We disagree and we consider the results have been presented following the objectives of the study.

Table S2: few models were below dAIC<2 and thus model averaging should be conduced I think for these Sandwich tern models.

The Authors: We agree and we mentioned that some models had similar AIC values although they include extra variables. However in comparing nested models these variables act as a “pretending varaibles” (see Anderson, D.R. 2008. Model Based Inference in the Life Science: a primer on evidence, ed Springer pg 65). They do not add any explanatory power (a small drop in deviance), but, because nested, they necessarily result in a similar AIC value. We wrote that:

“Models 2 to 6 including the additive effect of the interspecific competition exerted by Yellow-legged gulls, seawater turbidity, or the statistical interaction with the 3rd quartile winds, had similar explanatory power to Model 1 (i.e. ∆AIC values < 2; Table 2 and S2 Table). However, both the interaction and the additive terms in models 2 and 5 act as pretending variables. Pretending variables occur when after adding a new variable in a model, a ΔAIC~2 is obtained but, the deviance does not decrease. [51] and should not be considered further (Appendix B in [52]).”

Discussion

In my opinion the Discussion is too focused on the results and studied species, and therefore failed to generalize over a broader range of species, systems and life histories. There is also a lack of comparisons with studies that also investigated changed in egg volume such as Barrett et al. 2012, Bennett et al. 2017, Tomita et al. 2009, Verhoeven et al. 2019.

The Authors: We agree. Our first goal was to investigate the role of large-scale and local variables as linear predictors of interannual variations in egg volume. We are aware that the literature on egg size is vaste but we wanted to concentrate on these predictors.

These results are only correlative, could the authors think and suggest ways / experiments to answer this drawback?

The Authors: This is a nice and important question. Indeed our study is correlative and it is not possible to provide long-term experimental data in these species. In same cases, for example, the colony was made by thousands of breeding pairs. Indeed, most studies assessing the effect of large-scale climatic indices on ecological parameters are based on correlations as the mechanisms behind these processes are often too complex to be assessed empirically. In our study, we first considered empirical studies explaining the most elemental factors influencing these mechanisms (influence of food intake on egg volume, the influence of waves and visibility on foraging success etc.) and then, we assumed that this could be transferred to a broader spatial scale. For example, it is known that the egg volume partially depends on food intake. On the other hand, it is quite clear that food intake must strongly depend on two key factors: food abundance and foraging conditions. So, it makes sense to think that foraging conditions influence food availability to seabirds by modulating its detectability and accessibility. Several studies have measured empirically the influence of foraging conditions (wave height and/or visibility) on seabirds foraging success. Large-scale experiments at population level are opportunistic (e.g. a fishing moratory or the closure of a local open air landfill) and cannot be conducted on a long-term. It would certainly be interesting to have more experimental data.

Data

No readme.txt, and no codes, which make the analyses hardly reproducible. For example are the variables already scaled? What are the units of each variable? What family and link function were used for the GLMs? How models within models selection table were chosen?

The Authors: We agree. Now the original R-scripts with all the analyses are fully available.

Thank you on behalf of the authors for helping to improve this study.

References mentioned in this review

Barrett, R., Nilsen, E. & Anker-Nilssen, T. (2012). Long-term decline in egg size of Atlantic puffins Fratercula arctica is related to changes in forage fish stocks and climate conditions. Mar. Ecol. Prog. Ser., 457, 1–10.

Bennett, J.L., Jamieson, E.G., Ronconi, R.A. & Wong, S.N.P. (2017). Variability in egg size and population declines of Herring Gulls in relation to fisheries and climate conditions. ACE, 12, art16.

Catry, P., Lemos, R.T., Brickle, P., Phillips, R.A., Matias, R. & Granadeiro, J.P. (2013). Predicting the distribution of a threatened albatross: The importance of competition, fisheries and annual variability. Progress in Oceanography, 110, 1–10.

Grömping, U. (2006). Relative Importance for Linear Regression in R : The Package relaimpo. J. Stat. Soft., 17.

Harris, M., Anker-Nilssen, T., McCleery, R., Erikstad, K., Shaw, D. & Grosbois, V. (2005). Effect of wintering area and climate on the survival of adult Atlantic puffins Fratercula arctica in the eastern Atlantic. Mar. Ecol. Prog. Ser., 297, 283–296.

Kvalnes, T., Ringsby, T.H., Jensen, H. & Sæther, B.-E. (2013). Correlates of egg size variation in a population of house sparrow Passer domesticus. Oecologia, 171, 391–402.

Matthiopoulos, J. (2003). The use of space by animals as a function of accessibility and preference. Ecological Modelling, 159, 239–268.

Tomita, N., Niizuma, Y., Takagi, M., Ito, M. & Watanuki, Y. (2009). Effect of interannual variations in sea-surface temperature on egg-laying parameters of black-tailed gulls (Larus crassirostris) at Teuri Island, Japan. Ecol Res, 24, 157–162.

Verhoeven, M.A., Loonstra, A.H.J., McBride, A.D., Tinbergen, J.M., Kentie, R., Hooijmeijer, J.C.E.W., et al. (2020). Variation in Egg Size of Black-Tailed Godwits. Ardea, 107, 291.

Wakefield, E.D., Phillips, R.A., Trathan, P.N., Arata, J., Gales, R., Huin, N., et al. (2011). Habitat preference, accessibility, and competition limit the global distribution of breeding Black-browed Albatrosses. Ecological Monographs, 81, 141–167.

Weimerskirch, H. (2007). Are seabirds foraging for unpredictable resources? Deep Sea Research Part II: Topical Studies in Oceanography, 54, 211–223.

7. PLOS authors have the option to publish the peer review history of their article (what does this mean?). If published, this will include your full peer review and any attached files.

Do you want your identity to be public for this peer review? For information about this choice, including consent withdrawal, please see our Privacy Policy.

Reviewer #3: No

Reviewer #4: No

---

## [Editor Report · Decision Letter 2]

15 Aug 2022

It’s not all abundance: detectability and accessibility to food also explain breeding investment in long-lived marine animals

PONE-D-21-28485R2

Dear Dr. Real,

We’re pleased to inform you that your manuscript has been judged scientifically suitable for publication and will be formally accepted for publication once it meets all outstanding technical requirements.

Kind regards,

Vitor Hugo Rodrigues Paiva, Ph.D.

Academic Editor

PLOS ONE
---

## [Editor Report · Acceptance letter]

22 Aug 2022

PONE-D-21-28485R2

It’s not all abundance: detectability and accessibility of food also explain breeding investment in long-lived marine animals

Dear Dr. Real:

I'm pleased to inform you that your manuscript has been deemed suitable for publication in PLOS ONE. Congratulations! Your manuscript is now with our production department.

Kind regards,

on behalf of

Dr. Vitor Hugo Rodrigues Paiva

Academic Editor

PLOS ONE